# Influenza virus genome reaches the plasma membrane via a modified endoplasmic reticulum and Rab11-dependent vesicles

Isabel Fernández de Castro Martin[1], Guillaume Fournier[2,3,4], Martin Sachse[5], Javier Pizarro-Cerda[6,7,8,9,10], Cristina Risco[1] & Nadia Naffakh[2,3,4]

Transport of neo-synthesized influenza A virus (IAV) viral ribonucleoproteins (vRNPs) from the nucleus to the plasma membrane involves Rab 11 but the precise mechanism remains poorly understood. We used metal-tagging and immunolabeling to visualize viral proteins and cellular endomembrane markers by electron microscopy of IAV-infected cells. Unexpectedly, we provide evidence that the vRNP components and the Rab11 protein are present at the membrane of a modified, tubulated endoplasmic reticulum (ER) that extends all throughout the cell, and on irregularly coated vesicles (ICVs). Some ICVs are found very close to the ER and to the plasma membrane. ICV formation is observed only in infected cells and requires an active Rab11 GTPase. Against the currently accepted model in which vRNPs are carried onto Rab11-positive recycling endosomes across the cytoplasm, our findings reveal that the endomembrane organelle that is primarily involved in the transport of vRNPs is the ER.

[1] Centro Nacional de Biotecnologia, CNB-CSIC, Department of Macromolecular Structures, Cell Structure Lab, 28049 Madrid, Spain. [2] Département de Virologie, Institut Pasteur, Unité de Génétique Moléculaire des Virus à ARN, 75015 Paris, France. [3] CNRS, UMR3569, 75015 Paris, France. [4] Université Paris Diderot, Sorbonne Paris Cité, EA302, 75015 Paris, France. [5] Center for Innovation and Technological Research, Ultrastructural Bio-imaging, Institut Pasteur, 75015 Paris, France. [6] Département de Biologie Cellulaire et Infection, Institut Pasteur, Unité des Interactions Bactéries-Cellules, 75015 Paris, France. [7] INSERM, U220 8. INRA, USC 2020, 75015 Paris, France. [8] Unité de Recherche Yersinia, Institut Pasteur, 75015 Paris, France. [9] Centre National de Référence 'Peste et autres Yersinioses', 75015 Paris, France. [10] Centre Collaborateur OMS de Référence et Recherche 'Yersinioses', Institut Pasteur, Unité des Interactions Bactéries-Cellules, 75015 Paris, France. Isabel Fernández de Castro Martin and Guillaume Fournier contributed equally to this work. Correspondence and requests for materials should be addressed to C.R. (email: crisco@cnb.csic.es) or to N.N. (email: nadia.naffakh@pasteur.fr)

The cellular cytoskeleton and endomembrane system are exploited by viruses to promote multiple steps of the infection cycle, including viral entry, genome replication, intracellular trafficking of viral components, viral assembly, and egress. Studies on animal viruses have revealed different requirements for the microtubules and/or actin filaments and their respective motor proteins, dynein/dynactin and kinesin for microtubules and myosin for actin filaments[1]. They also revealed very diverse mechanisms of interaction between viruses and the highly dynamic organelles of the endomembrane system, such as the nuclear membrane, endoplasmic reticulum (ER), Golgi apparatus, endosomes, and vesicles. Best documented are the subversion of ER or Golgi membranes by positive-strand RNA viruses to generate viral factories, i.e., organelle-like structures in which viruses replicate[2,3], and the subversion of the endocytic recycling compartment (ERC) by negative-strand RNA viruses to promote the intra-cytoplasmic transport of neo-synthesized viral ribonucleoproteins (vRNPs)[4–6].

Unlike most RNA viruses, influenza A viruses (IAV) replicate in the nucleus of infected cells. Recent findings have improved our understanding of how their segmented negative-strand RNA genome, encapsidated into vRNPs, is transported into the nucleus to be transcribed and replicated, and how neo-synthesized vRNPs are exported from the nucleus and transported across the cytoplasm to the sites of viral budding at plasma membrane[7,8]. Infectious virions contain eight vRNPs, each consisting of a genomic RNA segment associated with nucleoprotein (NP) oligomers and with a copy of the hetero-trimeric PB1-PB2-PA polymerase[9]. Upon attachment to the host cell, IAV are internalized by either receptor-mediated endocytosis or macropinocytosis, after which they localize to early endosomes. Upon acidification of the endosomes and the subsequent fusion between viral and endosomal membranes, the vRNPs are released in the cytoplasm, transported into the nucleus through binding to cellular α/β-importins, and serve as templates for transcription and replication of the viral genome[7].

The nuclear export of progeny vRNPs is mediated by the CRM1-dependent pathway. Their connection to CRM1 is thought to be accomplished mainly through the formation of a daisy-chain involving the viral matrix protein (M1) and nuclear export protein (NEP/NS2)[10]. It is still controversial whether the vRNPs are exported from the nucleus individually, or as sub-bundles consisting of more than one but fewer than eight vRNA segments[11,12]. Upon nuclear export, vRNPs can be seen by immunofluorescence to accumulate transiently in a perinuclear region close to the microtubule organizing center (MTOC) and the associated ERC, characterized by the presence of Rab11 GTPases[13–15]. The cellular Y-box binding protein-1 (YB-1) and Human immunodeficiency virus Rev Binding protein (HRB) may facilitate the accumulation of vRNPs near to the MTOC[16,17]. There are many evidence that Rab11 is involved in vRNP trafficking[11–14,18]. The current view is that it mediates the docking of single vRNPs or vRNP sub-bundles to recycling endosomes in the vicinity of the MTOC through direct or indirect interaction of its active GTP bound form with the viral polymerase[13,19]. Thus, it has been proposed that recycling endosomes would carry the vRNPs across the cytoplasm. The infected cells show alterations in Rab11 distribution and recycling pathway efficiency, which is likely related to the fact that vRNPs hinder Rab11 binding to its effectors, the Rab11-family-interacting-proteins (FIPs)[20]. At late time points in infection, accumulation spots of vRNPs and Rab11 can be observed by immunofluorescence beneath the plasma membrane, from which vRNPs, but not Rab11, reach the plasma membrane and become incorporated into budding virions[14]. Most virions incorporate a full genome bundle consisting of eight distinct vRNPs, which get assembled together with the viral glycoproteins and matrix proteins in lipid raft-containing membrane domains[21]. Segment-specific *cis*-acting packaging signals have been defined on each vRNA, and there is growing evidence that they are involved in inter-segment RNA–RNA interactions that are guiding the genome packaging process[22,23].

Despite active research in the area, several aspects of vRNP transport from the nucleus to the plasma membrane remain poorly understood. It is unclear which cytoskeletal components and motor proteins are driving vRNP transport. Indeed, only a moderate decrease of the production of infectious virions is observed upon microtubule disruption or depolymerization of actin filaments[15,24–26], suggesting that both the microtubule network and actin filaments contribute jointly to a complex vRNP transport process, and/or that other mechanisms of transport are involved. It is unclear when and where vRNP sub-bundles are formed, as the resolution of single-molecule FISH analysis is not sufficient to conclude that the observed co-localization of distinct genomic RNA segments reflects a physical interaction[11,12]. Finally, the spatial distribution of Rab11 vesicles carrying vRNPs is unknown, and the mechanism underlying the dissociation of vRNPs from Rab11 vesicles prior to packaging remains to be demonstrated.

Here we unexpectedly identify that the ER is subverted by IAVs to promote the transport of neo-synthesized vRNPs from the nucleus to the plasma membrane. We show that IAV infection induces a major remodeling of the ER around the microtubule organizing center (MTOC) and all throughout the cell, as well as the formation of new organelles consisting of irregularly coated vesicles (ICVs). We demonstrate that vRNP components and the Rab11 protein are present on the modified ER and on ICVs, and that ICVs are distinct from recycling endosomes. Some ICVs are found very close to the ER and to the plasma membrane, and we provide evidence that Rab11 function is required for ICV formation. Overall our data strongly support a model in which (i) the modified ER is the first station of vRNPs after their exit from the nucleus, (ii) the ER is involved in the Rab11-dependent biogenesis of ICVs displaying Rab11 and vRNPs, and (iii) ICVs then serve as the transport organelle for vRNPs from the ER to the plasma membrane.

## Results

**IAV infection induces the recruitment of rough ER around the MTOC.** Several independent studies have shown by immunofluorescence microscopy an accumulation of IAV vRNPs in a perinuclear region close to the MTOC from where they are transported to the plasma membrane in a Rab11-dependent manner[13,18,27]. To visualize this event by both light and electron microscopy we generated an A549 cell line constitutively expressing an HA-MT-tagged Rab11A protein following the strategy described in Supplementary Fig. 1A. This allowed us to select clones that do not overexpress high amounts of the tagged Rab11 constructs. The western-blot analysis confirmed that expression of tagged Rab11 in these cells is in the same range as for the endogenous Rab11 protein and is unchanged upon infection with the A/WSN/33 (WSN) virus (Supplementary Fig. 1B). Immunofluorescence and confocal microscopy showed that HA-MT-Rab11 and the IAV nucleoprotein (NP) co-exist in a perinuclear region, up to 8 h post infection (hpi) (Fig. 1a, b, Supplementary Fig. 1C). Transmission electron microscopy (TEM) of oriented serial sections showed that the perinuclear region corresponding to the MTOC was drastically modified upon IAV infection (Fig. 1c, d). In a mock-infected cell (Fig. 1c), the MTOC is typically surrounded by mitochondria, rough ER (further referred to as ER) and small vesicles. However, in IAV-

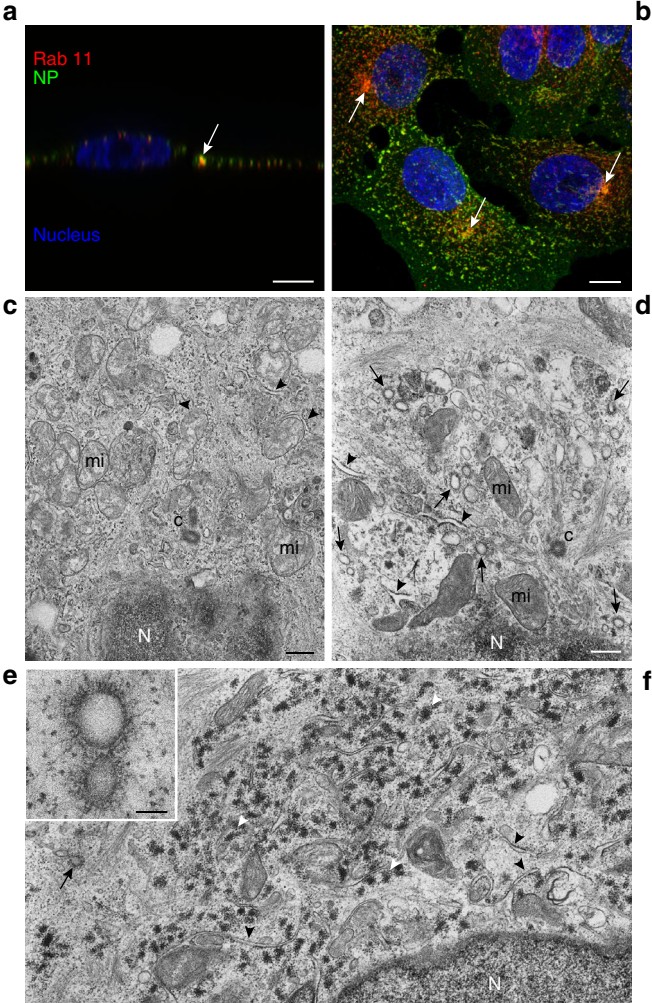

**Fig. 1** WSN-infected A549 cells recruit rough ER cisternae and a new type of organelle around the MTOC. **a**, **b** Immunofluorescence and confocal microscopy images (lateral and frontal merge, respectively) of A549 cells expressing HA-tagged Rab11 and infected with the WSN virus (8 hpi). Rab11 (red) and viral NP (green) co-exist in discreet spots near the nucleus (arrows). (**c**) Ultrathin-section of a mock-infected A549 cell as visualized by transmission electron microscopy (TEM). The MTOC area near the nucleus is shown. Mitochondria (mi) surround the centrioles (c) near the nucleus (N). **d** Equivalent MTOC region in a WSN-infected A549 cell at 8 hpi showing swollen rough ER cisternae (arrowheads) and numerous irregularly coated vesicles (ICVs, arrows) that are a new type of organelle. **e** Close-up of a pair of ICVs. **f** General view of perinuclear region in a WSN-infected A549 cell at 8 hpi showing modified rough ER (black arrowheads), groups of ribosomes (white arrowheads), and an ICV (arrow). Scale bars, 10 μm **a**, **b**; 0.5 μm **c**, **d**, **f**; 100 nm **e**

infected cells at 8 hpi (Fig. 1d) the MTOC is surrounded by swollen rough ER (arrowheads) and numerous vesicles with an electron lucent lumen and an irregular coat on their cytoplasmic side (arrows). This coat consists of electron dense filaments, which show on thin sections a variable length and different spacing toward their neighbors (Fig. 1e). The morphology of the irregular coat differs from the well described cellular coats involved in transport between organelles, such as clathrin, caveolin, and COP coats (Supplementary Fig. 2), all of them displaying a regular organization of their coat. These irregularly coated vesicles (ICVs) were found in IAV-infected cells exclusively.

**Modification of the ER upon IAV infection extends throughout the cytoplasm.** Besides the modification of the MTOC area, we observed in WSN-infected A549 cells at 8 hpi a general transformation of ER. Long tubulated ER elements expanded from the nucleus to the plasma membrane, forming numerous three-way junctions (Fig. 2a), unlike what observed in mock-infected cells (Fig. 2b). Strikingly, ER membranes showed a partial or total loss of ribosomes, and groups of ribosomes formed aggregates in the cytoplasm next to ER cisternae (Fig. 1f, Supplementary Fig. 3A, white arrowheads). ICVs were observed in contact or close to swollen ER cisternae at distance from the MTOC (Supplementary Fig. 3A–C), although less frequently than in the vicinity of the MTOC (as quantified on $n = 10$ cells, Supplementary Table 1; $p = 0.002$, Wilcoxon matched-pairs signed-rank test). The observed phenotypes (accumulation of ICVs near the MTOC, ER remodeling and ribosome aggregation) were analyzed in $n = 18$ infected cells that were selected randomly. All three phenotypes were present simultaneously in all cells examined, therefore the three phenotypes induced by IAV infection are potentially related to each other. Importantly, these observations were not restricted to the human airway epithelial cells A549. We observed ER tubulation, aggregation of detached ribosomes together with ICVs also in IAV-infected Calu-3 cells (Supplementary Fig. 4A–C). Interestingly, in Calu-3 cells close-ups revealed that the filamentous elements of ICVs are very similar to filaments found on the cytoplasmic side of the plasma membrane in virus budding areas (Supplementary Fig. 4D, arrowheads).

To gain insight into the different stages of ER remodeling caused by IAV infection we vizualised IAV infection by electron microscopy at different time points (Fig. 3). The characteristic morphology of ER cisternae found in mock-infected cells (Fig. 3a) changed at 4 hpi when tubulation of ER started (Fig. 3b, black arrowheads). Dense aggregates inside the nucleus were also characteristic of this time point (Fig. 3b, white arrowheads). At 6 hpi tubular ER membranes expanding from the nuclear envelope to the plasma membrane were seen in infected cells (Fig. 3c, d). Whereas ER cisternae close to the cell surface were oriented parallel to the plasma membrane in mock-infected cells, they were perpendicular to it in infected cells (Supplementary Fig. 5A, B, respectively). ICVs are often present in contact to the remodeled, tubular ER (Fig. 3d, arrows). About 10% of them show within their lumen smaller electron-dense vesicles (Supplementary Fig. 5C) similar to those observed in some ER cisternae (Supplementary Fig. 5B, blue arrow). Confocal microscopy further demonstrated that the tubulation of the ER in IAV-infected cells is not a local, subcellular event but takes place throughout the cell (Fig. 3e, f).

We assessed the role of IAV membrane proteins in ER modification. We performed infections in A549 cells that had been pre-treated with siRNAs targeting the viral M2 ion channel. Almost 100% of the cells were both efficiently silenced and infected, as revealed by indirect immunofluorescence staining for M2 and NP at 8 hpi (Supplementary Fig. 6A). Upon TEM analysis, they showed the ER remodeling and ICVs characteristic of infection (Supplementary Fig. 6B). So did A549 cells infected with a recombinant IAV pseudotyped with the VSVG envelope glycoprotein, that expresses neither the HA nor the NA and can undergo multicycle replication (Supplementary Fig. 7). Taken together, our data demonstrate that viral transmembrane proteins are not requested for ER remodeling and ICV formation. Notably however, the average size of ICVs was larger in cells infected with the pseudotyped virus compared to the wild-type (61% compared to 10% ICVs showing a diameter ≥200 nm, respectively, as determined on 100 ICVs for each experimental condition).

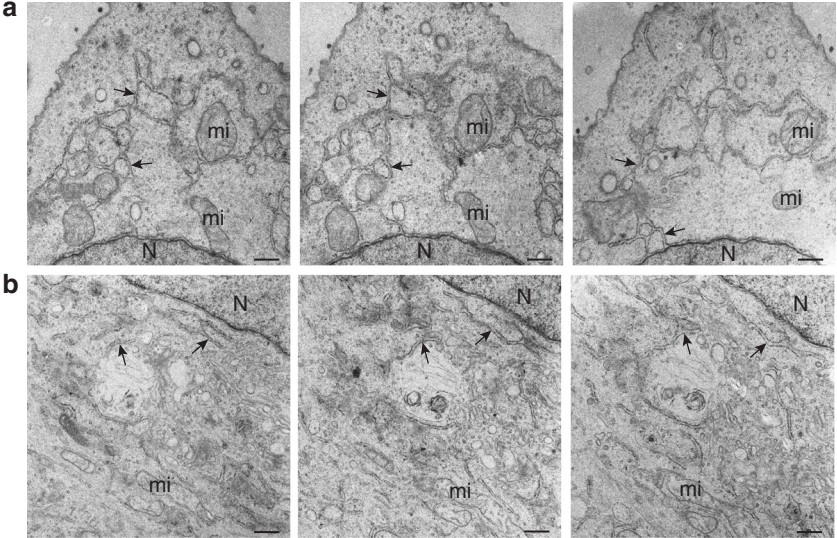

**Fig. 2** IAV induces a general remodeling of the ER. **a** Serial sections of a WSN-infected A549 cell at 8 hpi showing long ER elements (arrows) expanding from the nucleus to the cell periphery. **b** Serial sections of a mock-infected A549 cell. ER cisternae near the nucleus (N) are marked with arrows; mi, mitochondria. Scale bars, 0.5 μm

**vRNPs are recruited to the ER after their exit from the nucleus**. Based on the structural similarities between the ICVs' filamentous coats and the filaments localized underneath the plasma membrane in budding areas of viruses (Supplementary Fig. 4D), we hypothesized that ICVs might carry vRNPs. We measured the length of 137 randomly selected filaments from 16 ICVs (Supplementary Fig. 8A–D); it ranged from 26.6 to 112.5 nm, which was consistent with the 30–110 nm range described for the vRNPs[28,29]. To further assess our hypothesis, we imaged the transport of IAV vRNPs from the nucleus to the plasma membrane with metal-tagging transmission electron microscopy, or METTEM. This approach is a highly sensitive nanotechnology that has permitted identification of virus-induced organelles[30–32]. A recombinant WSN virus with the metal-binding protein metallothionein (MT) fused to the PB2 subunit of the viral polymerase was obtained (Supplementary Fig. 9). PB2 is a component of vRNPs and visualization of PB2 molecules in the cytoplasm reveals the location of vRNPs during their transport from the nucleus to the plasma membrane[24]. A549 cells infected with the PB2-MT virus were incubated shortly with gold salts in vivo before fixation, and incubated with silver salts after fixation. Atoms of silver nucleate around the gold nanoclusters built by MT in the MT-tagged protein molecules, therefore enlarging them and making them visible in stained samples. Labeling specificity was confirmed in cells infected with the wild-type WSN virus, incubated with gold salts and silver and processed as described above (Supplementary Fig. 10). After post-fixation and embedding, cells were sectioned, stained and studied by TEM to visualize the location of vRNPs in different cell compartments (Fig. 4). At 8 hpi, vRNPs were detected at the nuclear envelope (Fig. 4a) and at the remodeled ER and ICVs, that are particularly abundant around the MTOC (Fig. 4a–c). Signals revealed also the presence of vRNPs in nuclear pores (Fig. 4d, arrows), attached to filaments (Fig. 4d, e, white arrowheads), on ICVs far from the MTOC in virus budding areas (Fig. 4e, f, arrows) and in extracellular viruses (Fig. 4f, arrowhead). Quantification on $n = 29$ cells showed that vRNPs associated to the nuclear envelope, ER or ICVs are significantly more abundant than vRNPs found in the cytosol or associated to the filaments ($p < 0.0001$, Wilcoxon matched-pairs signed-rank test) (Supplementary Table 2). We hypothesize that a limited accessibility of the MT-tag when

vRNPs are densely packed could account for the fact that some virions, as well as some ICVs, were not labeled with gold-silver nanoclusters.

To confirm the presence of vRNPs on the ICVs and remodeled ER, we used a second, independent labeling approach: A549 cells were infected with the WSN virus and fixed for immuno-electron microscopy (immuno-EM) at 16 hpi. Thawed cryosections were double labeled for NP (10 nm) and PB2 (15 nm). In the nucleus and in virions, PB2 and NP co-localization was observed only occasionally (Supplementary Fig. 11A, D), indicating a limited sensitivity of the co-localization assay. ICVs of spherical or tubular aspects showed signal for both NP and PB2 (Supplementary Fig. 11B, C), strengthening our conclusion that they are coated with vRNPs. An additional double-labeling was performed for PDI (Protein Disulfide Isomerase, 5 nm gold) and NP (10 nm gold). NP labeling was found on ER cisternae, which are characterized by the presence of PDI (Supplementary Fig. 12, black and white arrows). By contrast, ICVs showed signal for NP as well but were devoid of label for PDI; the signal for NP present on ICVs was higher compared to the lower amount of label found on ER cisternae (Supplementary Fig. 12, arrowheads).

Taken all together, these results strongly suggest that after their exit of the nucleus through nuclear pores vRNPs are present on remodeled ER membranes and are even more abundant on ICVs, which could represent the transport organelles of vRNPs to the site of virus budding.

**Rab11 is present on the ER and Rab11 function is required for ICV formation**. Because previous works demonstrated that Rab11 is involved in vRNP trafficking, we wanted to study the precise localization of this protein in IAV-infected cells and its potential connection with the newly identified organelles, the ICVs. A549 cells were analyzed by immunofluorescence and confocal microscopy upon staining for Rab11 and PDI (red and green in Fig. 5, respectively). In infected cells, several Rab11-PDI co-localization spots were observed at 8 hpi, suggesting a proximity between Rab11-positive vesicles and the tubular ER elements (Fig. 5a, b, arrows). In mock-infected cells, no or fewer Rab11-PDI co-localization spots were observed (Fig. 5c, d). The HA-MT-tagged Rab11 expressing cells were then processed for

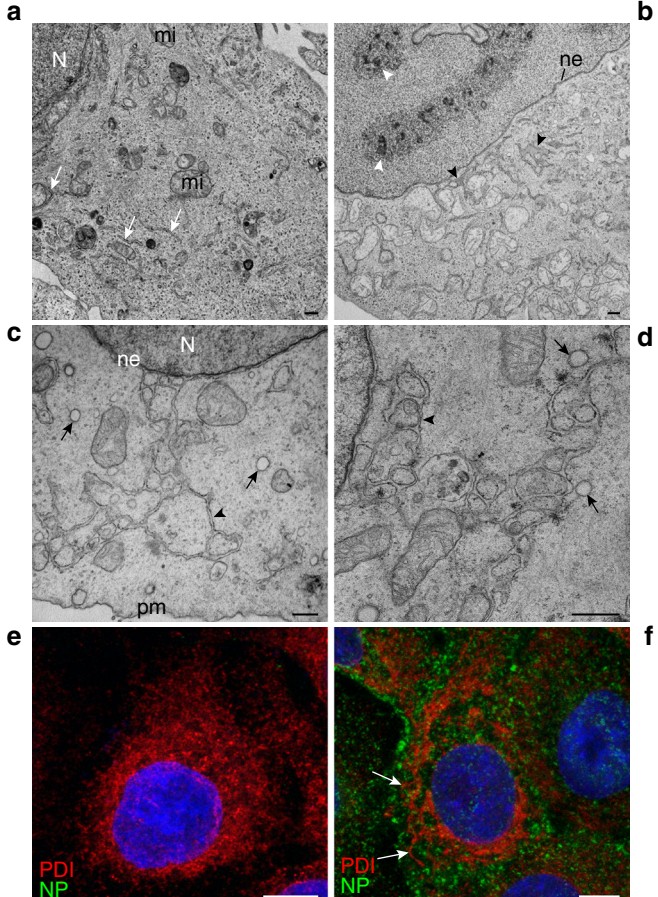

**Fig. 3** Time-course of influenza virus infection shows a progressive remodeling/tubulation of ER in A549 cells. **a** TEM of a mock-infected A549 cell. Mitochondria (mi) and rough endoplasmic reticulum (white arrows) have a random distribution. N, nucleus. **b** WSN-infected A549 cell at 4 hpi. The cell contains intranuclear dense structures (white arrowheads). Black arrowheads mark tubular, swollen ER membranes near the nuclear envelope (ne). **c**, **d** WSN-infected A549 cells at 6 hpi. Long ER-like tubular membranes (black arrowheads) expand from the nuclear envelope (ne) to the plasma membrane (pm). Arrows point to ICVs. **e**, **f** Immunofluorescence and confocal microscopy images showing viral NP (green) and the ER marker PDI (red) in mock- and WSN-infected A549 cells, respectively. Swollen/tubulated ER expands from the nucleus in WSN-infected cells at 8 hpi (arrows in **f**). Scale bars, 0.5 μm **a**–**d**; 10 μm **e**, **f**

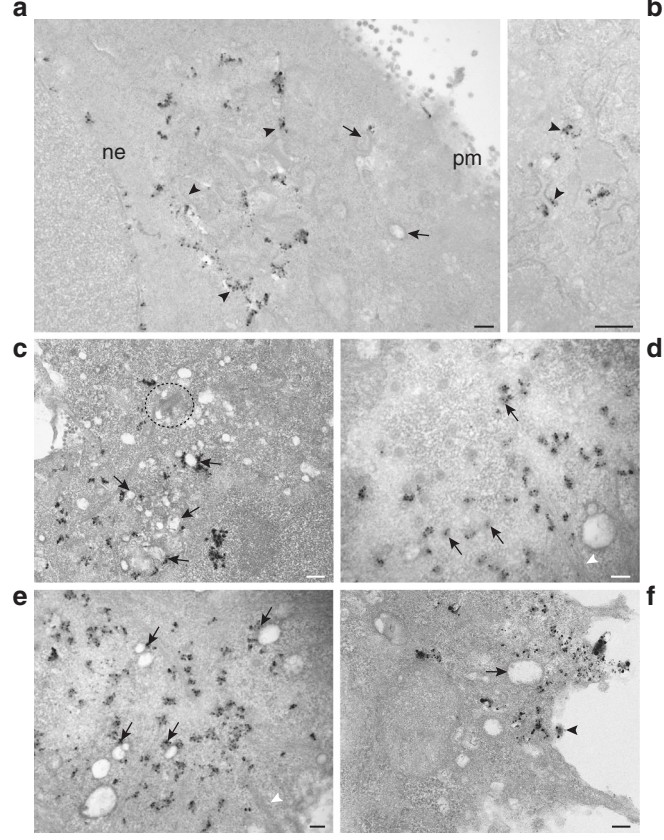

**Fig. 4** WSN vRNPs are detected in the ER and ICVs by METTEM. A549 cells infected with the recombinant virus WSN-PB2-MT were treated with gold salts at 8 hpi, fixed and incubated with silver to increase the diameter of gold nanoclusters built by the MT tag. Cells were post-fixed with osmium tetroxide, sectioned, stained and studied by TEM. **a** Cell showing labeled vRNPs in nuclear envelope (ne), in ER (arrowheads), ICVs (arrows) and close to plasma membrane (pm). **b** Higher magnification from a different cell showing labeled vRNPs in ER membranes (arrowhead). **c** Cell section through the MTOC showing label in ICVs (arrows) around the centriole (dashed circle). **d** Cell sectioned through the nuclear envelope showing vRNPs in nuclear pores (arrows) and close to filaments (arrowhead). **e** Signal associated with vRNPs in ICVs (arrows) and close to filaments (arrowheads). **f** Signal associated with vRNPs in a region close to plasma membrane. An ICV (arrow) and budding viruses (arrowhead) are labeled. Scale bars, 0.5 μm **a**–**c**; 200 nm **d**–**f**

METTEM following the same procedure described before for vRNP detection. In mock-infected cells, Rab11 was detected in vesicles and sporadically also on ER cisternae (Supplementary Fig. 13). In IAV-infected cells, a strong Rab11 signal was detected on some large portions of the remodeled ER (Fig. 6a, arrowheads) and on ICVs (Fig. 6a, arrows). Double immunogold labeling on thawed cryosections revealed the co-localization of Rab11 (5 nm gold) and NP/vRNPs (10 nm gold) in ICVs with their characteristic irregular electron dense coat (Fig. 7a) and in tubular ICVs (Fig. 7b, arrows). Only the presence of NP was detected in extracellular viral particles, that lacked signal for Rab11 (Fig. 7b). Double-labeling was then performed for Rab11 (5 nm gold) and the transferrin receptor (TfR, 10 nm gold), a marker of recycling endosomes[33]. Rab11 and TfR frequently co-localized to small vesicles with a morphology characteristic of recycling endosomes, notably an electron-dense lumen[34] (Supplementary Fig. 14A). In contrast, the vast majority of ICVs showed Rab11 signal but were devoid of TfR signal, which together with their electron lucent lumen clearly differentiated them from recycling endosomes (Supplementary Fig. 14B).

In the cells expressing the inactive, dominant-negative mutant form HA-MT-Rab11-S25N[35], viral-induced ER remodeling was observed. Unlike the active Rab11 protein, the mutant inactive Rab11 was not detected in the remodeled ER by METTEM and was mostly present on small smooth vesicles (Fig. 6b, arrows, insets, and Supplementary Table 3). No ICVs were detected in these cells, whereas a total of 143 ICVs were detected in cells expressing the wild-type Rab11 (n = 25 for each cell type, Supplementary Table 3). Similar observations were made upon double immunogold labeling of cells expressing Rab11-S25N, where NP signal was present on the ER and on small vesicles/tubules with no visible coat, and no ICVs were observed (Fig. 7c, d).

Our results clearly demonstrate that active Rab11 molecules are recruited to or redistributed on the ER, that the function of Rab11 is necessary for ICV formation, and that ICVs are new organelles distinct from recycling endosomes.

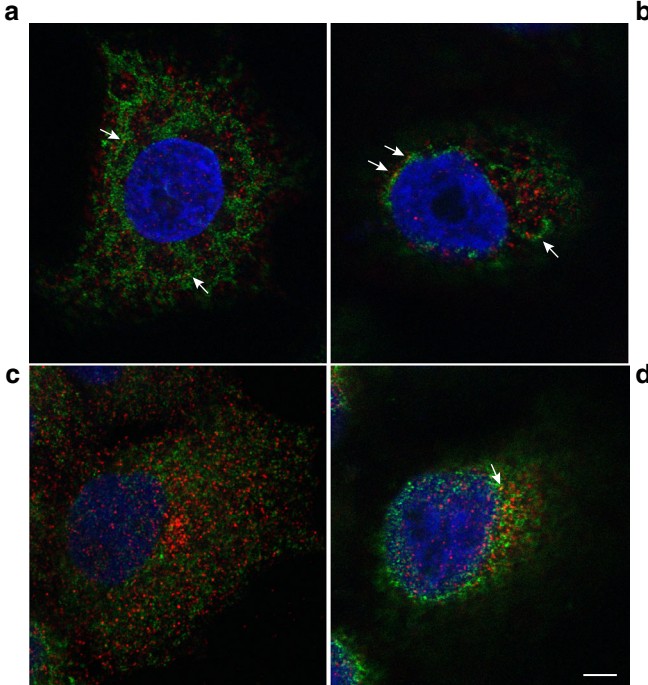

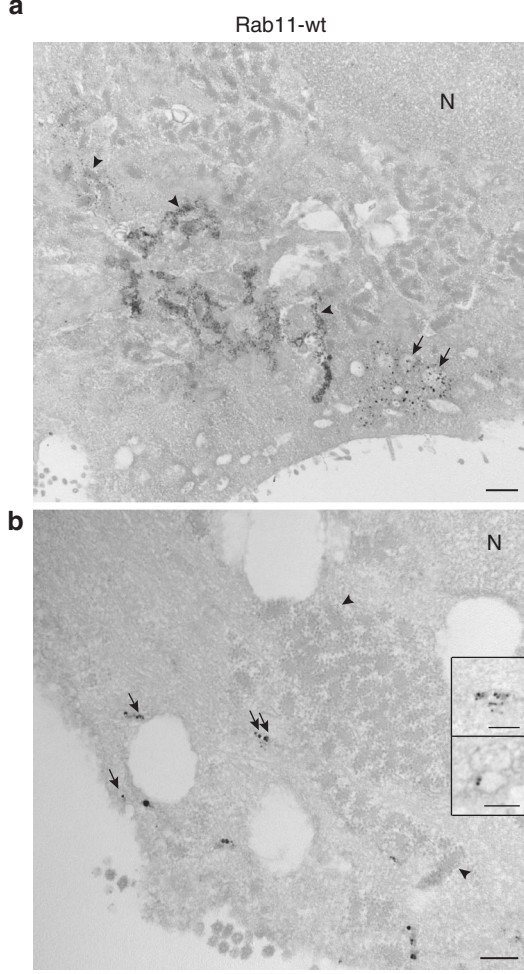

**Fig. 5** Localization of Rab11 and PDI in WSN-infected A549 cells. **a**, **b** Immunofluorescence and confocal microscopy images of two A549 cells (one optical section per cell) infected with the WSN virus and labeled at 8 hpi with antibodies specific for Rab11 (red) and the ER marker PDI (green). **c**, **d** Two optical sections of a control mock-infected cells. Arrows point to co-localization spots. Scale bar, 5 μm

**ICVs are found very close to the ER and to the plasma membrane**. To analyze the relation between the modified ER and ICVs and budding sites at the plasma membrane we studied IAV-infected A549 cells in three dimensions. TEM of serial sections, 3D reconstruction and image processing revealed a large membranous compartment consisting of tubular ER elements that expand from the nuclear envelope to the plasma membrane (Fig. 8a). Some ICVs were present in close vicinity to the ER and budding profiles from ER were found (Fig. 8b, Supplementary Fig. 3A–C). Enlarged images are presented in Supplementary Figs. 15, 16 to highlight the continuity between the remodeled ER membrane, still associated with groups of ribosomes and covered with vRNP-like filaments, and the membrane of budding ICVs. At the cell periphery, some ICVs were found very close to the plasma membrane (Fig. 8c–d). We quantified the number of ICVs that were close (<200 nm) or distant (>200 nm) to the PM in n = 25 cells at 8 hpi, making a distinction between ICVs with a dense irregular coat observed as single vesicles (Fig. 8b) or as pairs (Fig. 8c), and ICVs with a sparse irregular coat, always observed as single vesicles (Fig. 8d). As shown in Supplementary Table 4, ICVs with a sparse irregular coat were mainly observed close to the plasma membrane, whereas ICVs with a dense coat either single or in pairs where equally distributed. No fusion events of ICVs with the plasma membrane were observed. Given these observations, and the structural similarities between the ICVs' filamentous coats and the filaments localized underneath the plasma membrane in budding areas of viruses, we propose that when ICVs reach the plasma membrane vRNPs are directly transferred from ICVs to virion budding spots in a touch-and-go manner. vRNPs could also possibly be transferred to the plasma membrane via ER-plasma membrane contact sites.

**Fig. 6** Ultrastructural localization of HA-MT-Rab11 and HA-MT-Rab11-S25N by METTEM upon IAV infection. **a** Distribution of stably expressed HA-MT-Rab11 in WSN-infected A549 cells at 8 hpi. Cells were incubated with gold and silver before processing. Signals reveal the location of HA-MT-Rab11 in remodeled ER (arrowheads) and ICVs (arrows). N, nucleus. **b** Distribution of stably expressed HA-MT-Rab11-S25N in WSN-infected A549 cells at 8 hpi. Signals show the presence of HA-MT-Rab11-S25N in small vesicles (arrows). The arrowheads label remodeled ER with groups of ribosomes. No ICVs are detected. Insets are enlargements of labeled vesicles. The inset on the top corresponds to the vesicle marked with a double arrow in the middle of the mainfield. After irradiation in the microscope the silver particles decrease in size and the electron lucent center of the vesicle is better seen. The inset on the bottom corresponds to a labeled vesicle from another cell. Scale bars, 0.5 μm **a**; 200 nm **b**. 100 nm (insets)

**Purification of ICVs and characterization by western blot and TEM**. To further characterize ICVs, we performed subcellular fractionation analysis of mock-infected and IAV-infected cells at 14 hpi, using sucrose gradient ultracentrifugation. Five fractions were collected and were analyzed by western-blot and TEM. The viral NP and PB2 proteins were found to co-fractionate together with Rab11 in the heaviest fraction from IAV-infected cells (Fig. 9a, right panel, 60–40% sucrose), and ICVs were observed upon negative staining of an aliquot of this same fraction (Fig. 9b, black arrows). Individual vRNP-like structures were present on the surface of ICVs (Fig. 9b, white arrows), very similar to isolated vRNPs as previously visualized by negative staining[29]. These

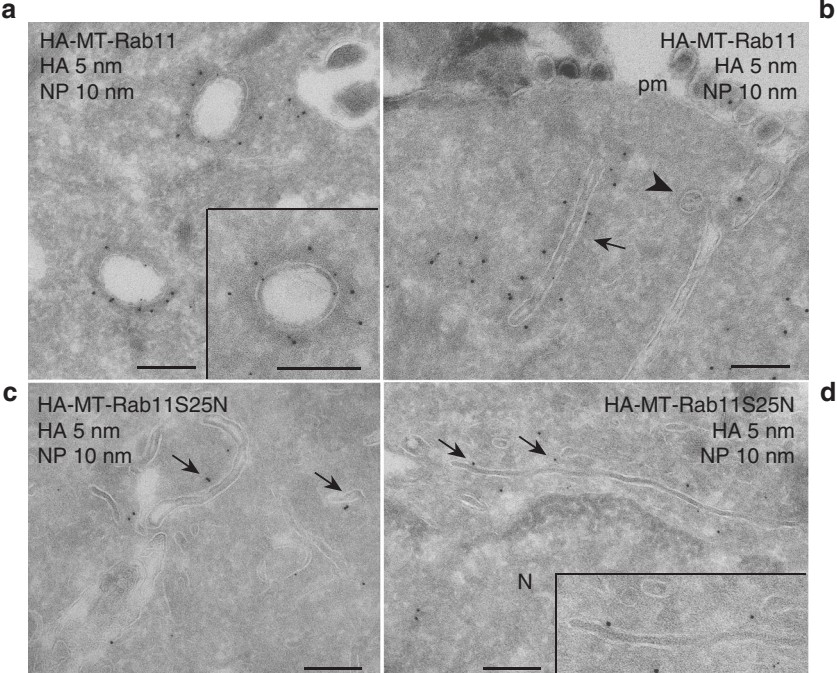

**Fig. 7** Rab11 co-localizes with NP on ICVs and is required for ICV formation. A549 cells expressing HA-MT-Rab11 **a**, **b** or HA-MT-Rab11-S25N **c**, **d** were infected with the WSN virus and fixed for immuno-EM at 16 hpi. Thawed cryosections were labeled for HA (5 nm gold) and NP (10 nm gold). **a** Two ICVs in the cytoplasm that are positive for both NP and Rab11. The insert shows in detail an ICV with the characteristic irregular electron dense coat and label for Rab11 and NP. **b** An area underneath the plasma membrane (pm) with viral particles outside the cell and a tubular ICV (arrow) close to it. Label for Rab11 and NP are found on the ICV. The irregular electron dense coat of the ICV can be clearly distinguished against the regular coat of clathrin, present on a clathrin-coated pit (arrowhead). **c**, **d** In cells expressing HA-MT-Rab11-S25N ICVs are absent. Instead, NP labeling is present on small vesicles and tubules (arrows). These tubules show no visible coat but often have an electron dense lumen. Scale bars, 200 nm

data strongly support our previous conclusions that ICVs are transport vesicles for vRNPs. Notably, the peak concentration of the Rab11 and TfR markers of recycling endosomes coincided in the 40–30% fraction for mock-infected cells (Fig. 9a, left panel), whereas in IAV-infected cells Rab11 and TfR concentration peaked in the 60–40% and 40–30% fractions, respectively (Fig. 9a, right panel). This difference, taken together with the fact that no ICVs were detected in the 40–30% fraction of infected cells upon negative staining, is in agreement with our Rab11-TfR double immunolabeling data (Supplementary Fig. 14A) and strengthens our conclusion that ICVs are distinct from recycling endosomes. The 60–40% fraction from infected cells was not enriched in clathrin, which was consistent with the very distinct morphology of ICVs and clathrin-coated vesicles (Supplementary Fig. 2). Finally, the pattern observed on western-blots for the ER markers calreticulin and PDI differed between mock-infected and IAV-infected cells (Fig. 9a). Not only calreticulin and PDI were relatively enriched in the 60–40% fraction from infected cells, but they were revealed as a doublet band instead of a single band in mock-infected cells, suggesting that they might undergo post-translational modifications upon IAV infection. For PDI as well as calreticulin, the band that was enriched in the 60–40% fraction from infected cells differed from the single band detected in mock-infected cells.

## Discussion

Our findings call into question the current view on how IAVs are subverting the ERC to promote the transport of neo-synthesized vRNPs. They reveal that the endomembrane organelle that is actually and primarily subverted by IAVs to transport vRNPs from the nuclear periphery to the plasma membrane is the ER. Instead of the model in which vRNPs are docked to Rab11-

positive recycling endosomes in the vicinity of the MTOC and are carried onto recycling endosomes across the cytoplasm[8], our data strongly suggest an alternative model (Fig. 10) in which (i) viral infection induces an extensive swelling and tubulation of the ER throughout the cell, from the nuclear envelope to the plasma membrane, (ii) the Rab11 protein and vRNPs newly exported from the nucleus are recruited to membranes of the remodeled ER, (iii) the modified ER is involved in the biogenesis of irregularly coated vesicles (ICVs) carrying Rab11 and vRNPs, which requires an active Rab11 protein, (iv) the newly formed ICVs ensures transport of vRNPs from the ER to the plasma membrane, and (v) vRNPs are released from ICVs and transferred to the plasma membrane in a touch-and-go process.

Numerous viruses are already known to induce modifications of the ER to promote their multiplication. However, unlike IAVs, most of them are replicating in the cytoplasm and they induce ER remodeling to generate viral replication and/or assembly factories[36]. The biogenesis of such factories is best described for positive-strand RNA viruses and involves either membrane invaginations towards the ER lumen, giving rise to spherules, or ER membrane exvaginations toward the cytosol, giving rise to single-membrane vesicles/tubules or double-membrane vesicles[3]. We show that IAV-infected cells present an extensive expansion of ER membranes, with the formation of single-membrane ER cisternae and ER tubules; instead of providing sites for replication factories, the expanded ER membranes serve as platforms for the gathering and transport of vRNPs. IAV-infected cell also present groups of detached and aggregated ribosomes, a feature possibly related to ER remodeling as ER-bound ribosomes were shown to play a role in ER morphology[37].

Why has such an extensive ER remodeling and association of ER membranes with vRNPs remained uncovered so far? To our

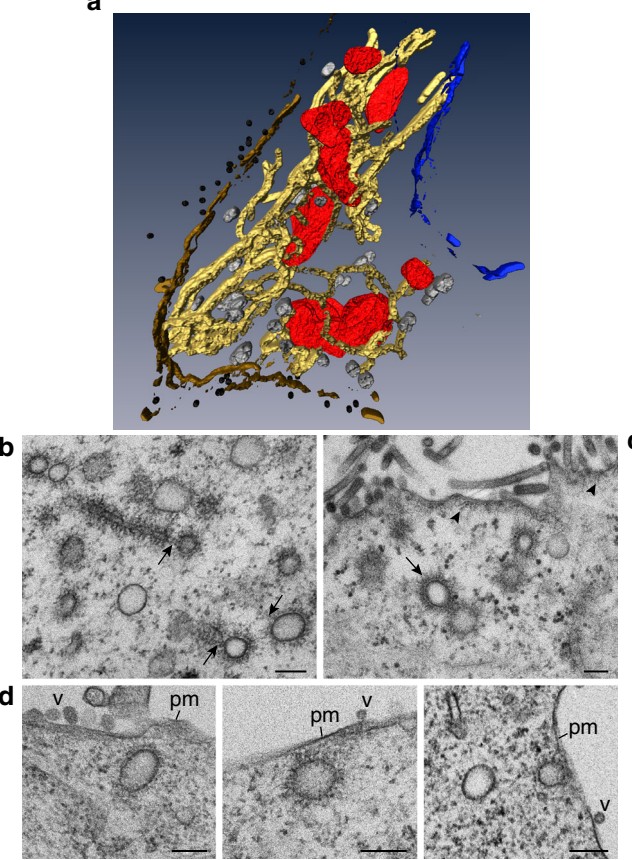

**Fig. 8** Remodeled ER and ICVs expands from nuclear envelope to plasma membrane in WSN-infected A549 cells. **a** WSN-induced membranous compartment at 8 hpi as visualized by TEM of serial sections, 3D reconstruction and image processing. Tubular ER membranes (yellow) with attached mitochondria (red) and ICVs (gray) contact with the nuclear envelope (blue) and the plasma membrane (brown). Viruses are shown in black. **b**–**d** Collection of 2D images showing the assembly of ICVs from modified ER membranes and the ultrastructure of ICVs at 8 hpi. **b** The membrane of ICVs is continuous with that of altered ER (arrows). **c** ICVs near a virus budding area. The irregular coat of ICVs (arrows) is very similar to the layer of short filaments under the plasma membrane (arrowheads). **d** Close-ups of ICVs very close to the plasma membrane (pm). V, viruses. Scale bars, 200 nm

knowledge, electron microscopy on flat-embedded IAV-infected cells and sectioned through the MTOC has not been performed before this study. Flat embedding allowed us to section single cells in their integrity and in a precise orientation, and to perform serial sectioning for 3D reconstruction. In addition, we used two optimized and complementary labeling methods to visualize viral and cellular proteins on electron microscopy sections. It is noteworthy here to consider the high sensitivity of the MT-tagging as it allows the detection of the protein of interest throughout the whole volume of the section. In contrast for immuno-EM, the labeling of the antibody is restricted to the epitopes on the surface of the section[38]. Importantly, the latter allows to sequentially label with different antibodies followed by protein A gold and to study on the same section the distribution of different proteins.

Another likely reason why ER involvement in the transport of vRNPs has not been detected earlier is that, although indirect immunofluorescence is a highly sensitive method, its dynamic range makes it unsuitable to reveal the co-existence of two markers that are present at low density in the same subcellular

compartment while one or both of them is present at high density in another compartment. Our immuno-EM data show that the density of Rab11 and NP is lower in the ER than in ICVs. This most likely accounts for the fact that the immunofluorescence signal for the ER-specific PDI marker in IAV-infected cells only punctually overlaps with the signals for Rab11 or the viral NP (Fig. 5a, b and Fig. 3f, respectively), whereas the immuno-fluorescence signals for Rab11 and NP, that are enriched in ICVs, do overlap (Fig. 1a, b).

Our data reinforce and further outlines the already known requirement of Rab11 for vRNP transport. However, it unexpectedly places the function of Rab11 not on recycling endosomes but instead on the ER. Indeed, using the dominant-negative Rab11-S25N mutant we demonstrate that an active Rab11 GTPase is required for the formation of ICVs, which might bud from the ER. The subcellular localization of Rab11 and the TfR in infected cells, as revealed by immuno-EM and by subcellular fractionation analysis, unequivocally demonstrate that ICVs are new organelles distinct from recycling endosomes. We detected Rab11, as well as the PB2 and NP components of vRNPs at ER and ICVs' membranes by immuno-EM, which is consistent with data from others showing that vRNPs can be co-immunoprecipitated with Rab11 and that the interaction with Rab11 is required for the targeting of vRNPs to endomembranes[13,18]. Our finding that Rab11 is only sporadically detected in association to ER membranes in mock-infected cells whereas a strong Rab11 signal is present at ER membranes in some but not all areas of the ER of IAV-infected cells, raises the question of the mechanism involved. Whether it involves organelle contacts between the ERC and ER compartments, a subcellular relocation of Rab11 from the ERC/trans-Golgi network to the ER, and/or a redistribution of Rab11 within the ER should be investigated in future studies, as well as the question of which viral and/or cellular factors are involved. Interestingly, we found that a particular form of the ER markers PDI and calreticulin accumulated in the fraction of IAV-infected cells that was enriched in ICVs, with an apparent molecular weight which differed from that detected in mock-infected cells. In the case of PDI, the difference in SDS-PAGE mobility could correspond to changes in the redox state of the protein[39], which was shown to regulate the equilibrium between its chaperone and its enzymatic activities, and to play an important role in the maintenance of ER homeostasis and the adaptative response to ER stress[40]. More generally, these observations suggest that IAV infection could trigger post-translational modifications of ER proteins, which in turn could mediate the remodeling of the endomembrane system and the formation of ICVs.

We hypothesized that the transmembrane viral proteins which transit through the ER, i.e., the HA and NA glycoproteins and the M2 ion channel, could be involved. However, we found that a recombinant IAV pseudotyped with the VSVG envelope glycoprotein, that expresses neither the HA nor the NA and can achieve multicycle replication, induces ER remodeling and ICV formation in A549 cells. This observation, in agreement with the fact that the vRNP trafficking process could be reconstituted by transiently expressing the vRNP components together with the NS1, NS2, M1, and M2 proteins and in the absence of HA and NA[13], indicates that the HA and NA glycoproteins are not strongly involved in endomembrane remodeling and vRNP trafficking. Although published observations suggest a possible interplay between the trafficking pathways of the M2 protein and vRNPs[17,41], we did not observe any significant effect of M2 depletion on ER remodeling and ICV formation. A complex cross-talk between several viral proteins, ER proteins, and the Rab11 GTPase is probably involved, which is a new area to be explored in the future.

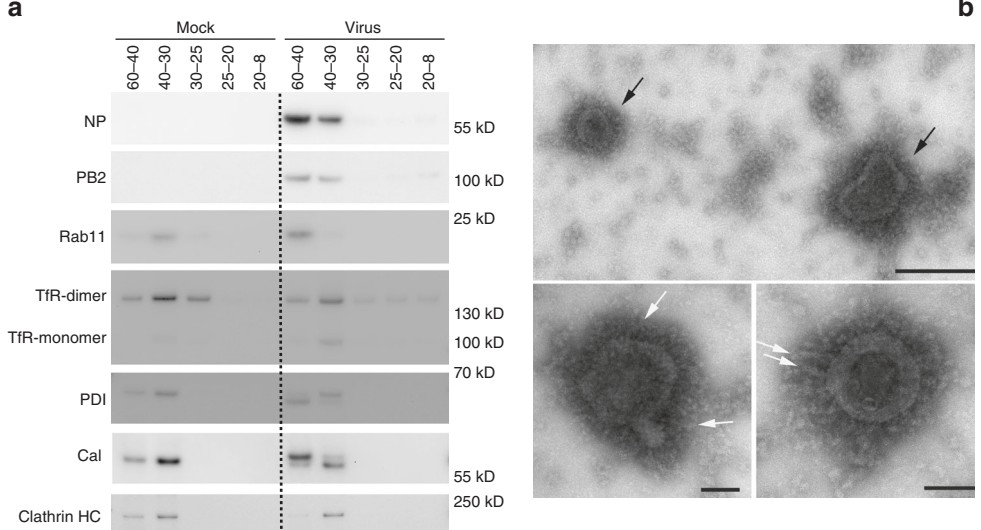

**Fig. 9** Characterization of purified ICVs. A549 cells were infected with the WSN virus and submitted to subcellular fractionation at 14 hpi. **a** Western-blot analysis of subcellular fractions of mock-infected (left panel) and IAV-infected (right panel) A549 cells, upon sucrose gradient ultracentrifugation. The same amount of total protein was analyzed for each fraction. TfR: transferrin receptor; PDI: protein disulfide isomerase; Cal: calreticulin; Clathrin HC: Clathrin Heavy Chain. For gel source data, Supplementary Fig. 17. **b** TEM analysis of an aliquot of fraction 60–40% from IAV-infected cells, upon fixation and negative staining. Black arrows: ICVs. White arrows: vRNPs. Scale bar: large panel 200 nm, small pannel 100 nm

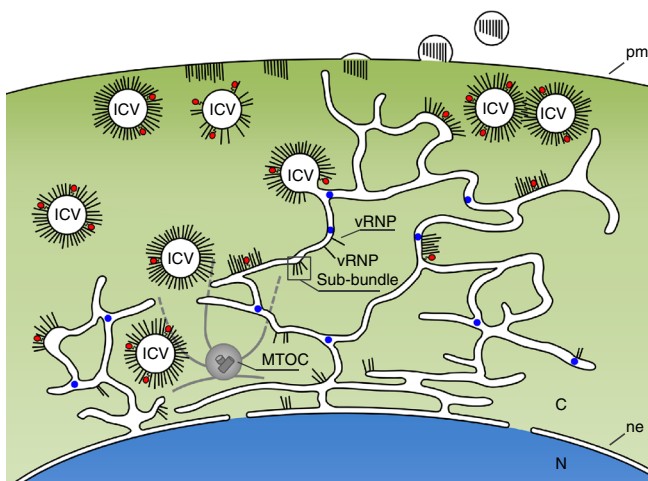

**Fig. 10** Model for the trafficking of vRNPs across the cytoplasm in an IAV-infected cell. The remodeled tubulo-vesicular ER, positive for the PDI marker (in blue) extends around the MTOC (in gray) and from the nuclear envelope (ne) to the plasma membrane (pm). After their exit from the nucleus, individual vRNPs and/or sub-bundles of vRNPs are targeted to the modified ER. ICVs loaded with vRNPs and with the Rab11 molecule (in red) might bud from the ER and ensure the transport of vRNPs to the plasma membrane. The frequently observed pairing of ICVs could favor RNA–RNA interactions among vRNPs and the progressive assembly of sets of 8 distinct vRNPs. vRNPs are released from ICVs and possibly transferred to the plasma membrane in a touch-and-go process

Our findings open new perspectives on what could be the motive force of vRNP transport. Indeed, the association of vRNPs with the membranes of a remodeled, expanded ER may provide an explanation for the only moderate effect of microtubule disruption or depolymerization of actin filaments on the production of infectious virions[8]. Actually, the expansion and dynamics of ER membranes by itself could contribute to the progression of vRNPs toward the plasma membrane, and be driven not only by a complex array of cytoskeleton and motor proteins, but also by

ER-localized membrane-shaping proteins and membrane-fusion mediating proteins[42]. In addition, the stream of the ER could drag the surrounding cytosol, as proposed for the transport of some plant viruses[43,44], and provide a motive force for the ICVs. Direct interactions between the ICVs and cytoskeletal filaments, as visualized in our electron microscopy samples, would represent an additional, complementary mechanism of motility.

Finally, our data provide the bases for new hypotheses about how vRNP sub-bundles are formed and how they are transferred to the plasma membrane. Remarkably, ICVs were frequently observed in pairs (Fig. 1e, Supplementary Fig. 3C), showing close contacts and interpenetration of the irregular coats formed by vRNPs. Given the known flexibility of vRNPs[45,46], this physical proximity could favor RNA−RNA interactions among vRNPs, and possibly vRNP flipping from one ICV to the neighboring one, leading to the progressive assembly of sub-bundles containing several and up to eight distinct vRNPs. Our images of ICVs very close to the plasma membrane, and plasma membrane areas lined with electron-dense coats very similar to the ICVs' coat, from which virions are budding, taken together with the fact that ICVs with a sparse coat were predominantly observed close to the plasma membrane, strongly suggest that vRNPs are released from ICVs and transferred to the plasma membrane in a touch-and-go process. The dissociation of vRNPs from ICVs most likely involves the GTPase activity of Rab11, assisted by a GAP (GTPase-activating protein) that remains to be identified, and interactions of vRNP components with viral proteins (e.g., M1) that are present at the plasma membrane[47]. Further elucidation of the late steps of IAV life cycle will require a combination of imaging, biochemical and genetic approaches, and might lead to the identification of novel therapeutic targets and the development of innovative antiviral strategies against influenza viruses.

## Methods

**Cells and viruses.** A549[48] and Calu3[49] human airway epithelial cells were obtained from Pr. Martin Schwemmle (Freiburg University) and Dr. Folker Schwalm (Marburg University), respectively. A549 and Calu-3 were grown in Dulbecco's modified Eagle's medium (DMEM) supplemented with 10% fetal calf serum (FCS). HEK293T[50] and MDCK[51] cells were obtained from Dr. Michel Perricaudet (CNRS-IGR, Villejuif, France) and from the National Reference Center

for Influenza Viruses at Institut Pasteur in Paris, respectively, and were grown in DMEM supplemented with 10% FCS and in Modified Eagle's Medium (MEM) supplemented with 5% FCS, respectively. A549 cells stably overexpressing a tagged Rab11A protein were obtained upon transfection with a pIRESpuro3- HA-MT-Rab11 or pIRESpuro3-HA-MT-Rab11-S25N plasmid using the JetPrime reagent (Polyplus Transfection), followed by the selection of cell clones resistant to 1 µg/mL puromycin (Invivogen). The M2 siRNAs (sense 5′-UUUCU-GAUAGGGCGUUUCGACCUCGG-3′ and antisense 5′-CCGAGGUCGAAACGC-CUAUCAGAAA-3′) were purchased from Invitrogen (stealth modified) and were transfected with Lipofectamine 2000 (Invitrogen), using 100 pmol siRNA and 3.5 µL of Lipofectamine (Invitrogen) per well in a 12-well format[52]. siRNA-treated A549 cells were infected at 18 h post transfection.

The recombinant A/WSN/33 (WSN) virus and its derivatives, were produced by co-transfecting a co-culture of HEK293T cells and MDCK cells with 8 reverse genetics plasmids for the synthesis of viral RNAs from the human PolII promoter and 4 plasmids for the expression of the PB1, PB2, PA, and NP viral proteins[53]. The PB2-MT and the HA(VSVG)NA(mCherry) viruses were produced by transfecting the pPolI-PB2-MT, and the combination of pPolI-HA(VSVG) and pPolI-NA(mCherry) plasmids, respectively, (described below) instead of the wild-type pPolI-PB2/HA/NA plasmids. Cultured cells were infected at a multiplicity of infection (moi) of 5 pfu per cell with the WSN and WSN-derived viruses, and were incubated with DMEM supplemented with 2% FCS for 4–16 h.

**Plasmids**. The pIRESpuro3-HA-MT-Rab11 plasmid was obtained by cloning sequentially into the pIRESpuro3 plasmid (Clontech) the Rab11 coding sequence amplified from the pDON207-Rab11 plasmid (human ORFeome resource, kindly provided by Y. Jacob) and the MT coding sequence, produced as a synthetic gene by GenScript. The reverse genetics plasmid pPolI-PB2-MT was obtained by replacing the Gluc1 sequence of pPolI-PB2-Gluc1 plasmid[54] with the MT sequence. The pPolI-HA(VSVG) and pPolI-NA(mCherry) were constructed by replacing the HA and NA coding sequences in the pPolI reverse genetics plasmids by the VSVG and mCherry coding sequences flanked by the 5′ and 3′ non-coding sequences derived from the HA and NA segment of the A/WSN/33 virus, respectively[55]. The VSVG and mCherry coding sequences were designed in silico to have the same codon usage as IAVs and were produced by GenScript. Directed mutagenesis was performed using the QuikChange II Site-Directed Mutagenesis Kit (Agilent) to obtain the pIRESpuro3-HA-MT-Rab11-S25N plasmid. All constructs were verified by Sanger sequencing. The sequences of the synthetic genes and the oligonucleotides used for amplification and sequencing can be provided upon request.

**Cell fractionation**. About $20 \times 10^6$ A549 cells were either mock-infected of infected with the A/WSN/33 virus at a moi of 5 pfu per cell. At 14 h post infection (hpi), cells were fractionated according to the protocol described in ref. [56]. Briefly, cells were washed and resuspensed in PBS-BSA 0.5%, and centrifuged at $300 \times g$ for 5 min. PBS-BSA 0.5% was exchanged for homogeneization buffer (8% sucrose in Imidazole 3 mM MgCl₂ 1 mM supplemented with EGTA 0.5 mM, gelatin 0.5% and complete protease inhibitors) and cells were centrifuged at $300 \times g$ for 10 min. The cells were then resuspended and mechanically disrupted in homogeneization buffer using a 25G5/8 needle. After centrifugation at $2000 \times g$ for 15 min, the post-nuclear fraction was collected, brought to 40% sucrose and loaded on top of a 60% sucrose cushion. A discontinuous 60/40%, 40/30%, 30/25%, 25/20%, and 20/8% sucrose gradient was prepared and ultra-centrifuged at $100,000 \times g$ for 1 h. The recovered fractions were adjusted at a final concentration of 10% sucrose, the protein concentration was determined using the Bradford reagent (Sigma) and 750 ng of each fraction were loaded on a NuPAGE 4–12% Tris-Glycine polyacrylamide gel (Thermo fisher) for immunoblot analysis.

**Immunoblots**. Total lysates from A549 cells stably overexpressing a tagged Rab11 protein were prepared in Laemmli buffer. Immunoblot membranes were incubated with primary antibodies directed against Rab11 (Invitrogen 71-5300, diluted 1/400), HA-tag (Biolegend 16B2, diluted 1/1000), NP (Kerafast EMS010, diluted 1/1000), PB2 (Thermo Fisher Scientific PA5-32220, diluted 1/5000), transferrin receptor (Thermo Fisher Scientific 13-6800, diluted 1/1000), PDI (Sigma P7372, diluted 1/1000), calreticulin (NovusBio NB600-101 diluted 1/1000), clathrin heavy chain (Thermo Scientific MA1-065, diluted 1/1000) and/or GAPDH (Thermo Fisher Scientific MA5-15738, diluted 1/2500), and revealed with secondary antibodies (GE Healthcare) and with the ECL 2 substrate (Pierce). The chemiluminescence signals were acquired using G-Box and the GeneSnap software (SynGene).

**Indirect immunofluorescence assays**. For indirect immunofluorescence assays cells were grown on glass coverslips, fixed with PBS-4% PFA, and permeabilized with PBS-0.1% Triton X-100 or with PBS-0.25% Saponin. The cells were incubated with the anti-NP (Meridian AA5H, diluted 1/500), anti-Rab11 (Invitrogen 71-5300, diluted 1/50), and/or anti-PDI antibodies (Sigma P7372, diluted 1/200), and then with DyeLight550-coupled anti-rabbit and/or DyeLight488 anti-mouse IgG secondary antibodies (Pierce, diluted 1/500) and/or AF488-coupled anti-HA-tag antibody (Molecular Probes, diluted 1/250) or AF488-coupled anti-M2 antibody

(Santa Cruz sc-32238, diluted 1/100). The samples were stained with Hoechst 33342 (Invitrogen, diluted to 1 µg/ml), mounted in ProLong Gold mounting medium (Life technologies P36930). For epifluorescence microscopy, images were acquired with upright Zeiss Axio Imager Z2 with HXP 120 lamp (at 50% level) using ×63 oil immersion objective lenses. Images were acquired with an AxioCam MRm camera and the ZEN blue 2012 software (Zeiss). For confocal microscopy images were acquired with a Leica TCS SP5 confocal multispectral microscope using a HCX PL APO 63.0 X/1.4 N.A. oil objective and LAS AF v.2.7.3 software (Leica Microsystems).

**Electron microscopy**. For ultrastructural studies, cells were grown on sterile Thermanox Plastic Coverslips (Nunc) prior to infection. Monolayers of mock- and IAV-infected cells were fixed at the indicated time points (1 h, at room temperature) in a mixture of 4% paraformaldehyde and 1% glutaraldehyde in HEPES (pH 7.4), post-fixed (1 h, 4 °C) with 1% osmium tetroxide and 0.8% potassium ferricyanide in water, dehydrated in 5 min steps, with increasing concentrations of acetone (50, 70, 90% and twice in 100%) at 4 °C and processed for flat embedding in the epoxy resin EML-812 (TAAB Laboratories). The cells were incubated overnight with a 1:1 mixture of acetone-resin at room temperature, infiltrated 8 h in pure resin and polymerized (48 h, 60 °C). Ultrathin (~70–80 nm) oriented serial sections were obtained with a UC6 ultramicrotome (Leica Microsystems), collected on uncoated 300-mesh copper grids (TAAB Laboratories), stained with saturated uranylacetate and lead citrate, and imaged by TEM. Images were acquired with a JEOL JEM 1011 electron microscope operating at 100 kv. More than 100 cells per condition were observed. At least three different resin blocks were sectionned for each experimental condition. Six independent experiments were performed on WSN-infected A549 cells.

To visualize metallothionein-tagged protein molecules (IAV PB2-MT, HA-MT-tagged Rab11, and HA-MT-Rab11-S25N), A549 cells were incubated in vivo shortly with 0.25 mM HAuCl₄ (Sigma-Aldrich) in DMEM for 30 min at 37 °C. This treatment builds gold nanoclusters in metallothionein-tagged proteins, allowing highly sensitive detection of protein molecules in cells[31,32]. The cells were then washed with DMEM, fixed in a mixture of 4% paraformaldehyde and 1% glutaraldehyde in HEPES, washed with deionized water and incubated 10 min with silver salts (HQ SILVER, Nanoprobes). After washing with deionized water, samples were post-fixed and embedded in epoxy resin as described above. Ultrathin sections were stained and imaged by TEM. For confirming labeling specificity, IAV-infected cells (lacking MT-tagged proteins) were incubated with gold and silver salts and processed as described above (Supplementary Fig. 10). At least three different resin blocks were sectionned for each experimental condition. Three independent experiments were performed on WSN-PB2-MT infected cells.

For immunolabeling on thawed cryosections, cells were fixed according to their time post infection with 2% PFA (EMS) and 0.1% glutaraldehyde in PHEM buffer, pH 7.2 (60 mM PIPES, 25 mM HEPES, 10 mM EGTA, 2 mM MgCl₂) for 2 h at RT. Afterwards free aldehydes groups were quenched with 50 mM NH₄Cl before cells were removed from the plastic with a rubber policeman and pelleted in a 1.5 ml Eppendorf tube. The pellet was embedded in 12% gelatin (TAAB Laboratories) in PBS and after solidification, cubes of 1 mm³ were cut and infiltrated overnight with 2.1 M sucrose in PBS at 4 °C. The cubes were mounted on metal pins and frozen in liquid nitrogen. Thin cryosections were cut at −120 °C using a FC6 (Leica microsystems) and picked up with a 1:1 mixture of 2% methylcellulose in H₂O and 2.1 M sucrose in PBS and placed after thawing on 200 mesh grids with a carbon coated Formvar film. Single and double labelings were perfomed using mouse anti-NP (Kerafast EMS101, diluted 1/200), mouse anti-TfR (Thermo Fisher Scientific 13-6800, diluted 1/100), rat anti-HA (clone 3F10, Roche 11867423001, diluted 1/200), mouse anti-PDI (gift from the late Stephen Fuller, diluted 1/20), and rabbit anti-PB2 (Thermo Fisher Scientific PA5-3220, diluted 1/20) primary antibodies, rabbit anti-mouse (Dako Z0259) and rabbit anti-rat (Epitomics 3030-1, R18-2) bridging antibodies, with Protein A gold (CMC-Utrecht), as described[57]. Briefly, remaining free aldehydes were quenched with 50 nm NH₄Cl in PBS and unspecific binding was blocked with 1% BSA in PBS. After the primary antibody a bridging antibody was used if necessary, followed by protein A gold. This complex was stabilized by 1% glutaraldehyde in PBS and sections were contrasted with 0.4% uranylacetate in 1.8% methylcellulose in water (single labeling) or further labeled with a second antibody before contrasting (double labeling). After labeling images were taken with a Tecnai G2 (FEI) run at 120 kV equipped with a Gatan US4000 (Gatan Inc.). At least three independent labelings were performed for each experimental condition.

For negative staining of subcellular fractions, aliquots of the fractions were inactivated by addition of ×10 concentrated fixative to a final concentration of 0.5% glutaraldehyde in ×0.1 PHEM buffer, pH 7.2 in the fraction. Before deposition of the suspension on carbon-coated 200 mesh copper palladium grids, the grids were glow discharged in a Quorum Q150R ES (Quorum). After adhesion for 3 min the grids were rinsed with water and stained with 2% aqueous uranylacetate solution and viewed in a Tecnai Spirit G2 electron microscope operated at 120 kV (FEI). Two independent negative stainings were performed.

**3D image reconstructions**. Consecutive ultrathin sections were collected on Formvar-coated copper slot grids (Taab Laboratories), stained, and imaged by TEM. A total of 8 series of 12 were selected and processed for 3D reconstruction[58].

Photographs of IAV-infected cells were taken at a nominal magnification of ×25,000. Plates were digitized as 8-bit images with a 3.39 nm final pixel size and a 600-dpi resolution with an EPSON perfection Photo V500 scanner. Digital images were aligned using selected tracers between two consecutive sections with the free editor for serial section microscopy Reconstruct[59] (http://synapses.clm.utexas.edu/tools/index.stm). Noise was reduced with three rounds of median filter, segmentation and 3D visualization using Amira software (http://amira.zib.de).

**Data availability**. The authors declare that the data supporting the findings of this study are available within the paper and its Supplementary Information.

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

## Acknowledgements

We thank Sandie Munier and Pierre Loap for contributing to the production of the WSN-PB2-MT virus, Sylvia Gutiérrez Erlandsson and Ana Oña Blanco for expert support with confocal microscopy, and Arnaud Echard and Sylvie van der Werf for helpful discussions. We thank the Imagopole-CITech, part of the FranceBioImaging infrastructure supported by the French National Research Agency (ANR-10-INSB-04-01), "Investments for the future". This research was supported by the EU FP7 PRE-DEMICS project (278433) (NN) and the research grant BIO2015-68758-R (MINECO-FEDER) from the Spanish Ministry of Economy, Industry and Competitiveness (CR). The Institut Carnot Pasteur Maladies Infectieuses and the PREDEMICS project supported G.F.

## Author contributions

I.F.d.C.M., G.F., M.S., J.P.-C., C.R., and N.N. designed and conducted the experiments. N.N., C.R., and M.S. wrote the paper with input from other authors.

## Additional information

**Competing interests:** The authors declare no competing financial interests.

