## [Peer Review File · Nature Communications]

Reviewers' comments:

Reviewer #1 (Remarks to the Author):

A new class of Rab11-dependent vesicles transports influenza virus genome from a modified endoplasmic reticulum to the plasma membrane
de Castro Martin et al. Nature Communications

Influenza A virus (IAV) vRNPs require Rab11, a recycling endosomal GTPase, for egress toward the plasma membrane. After replication in the nucleus, progeny vRNPs are nuclear exported by a CRM1-dependent pathway dependent on matrix protein M1 and nuclear export protein (NEP), and facilitated by cellular factors such as YB-1. After initial concentration at the MTOC, it has been suggested that vRNP egress occurs by piggy-backing on Rab11-positive recycling endosomes that traffic outward to the cell periphery along microtubules. Rab11-depletion or dominant-negative expression reduces the release of infectious progeny virions (Eisfeld et al., 2015; Lakdawala et al., 2016).

In this study, de Castro-Martin et al. show using conventional EM, cryo EM and metal-tagging TEM (METTEM) that progeny vRNPs are found on the surface of irregularly coated vesicles (ICVs), endomembrane organelles that bud from the ER and touch the plasma membrane (PM). Infected cells produce new organelles consisting of ICVs on which progeny vRNPs are present but protein disulphide-isomerase (PDI) is not. They propose an alternative model of vRNP egress: (i) vRNPs locate to the ER following export, (ii) ICVs displaying Rab11/vRNPs bud from the ER in a Rab11-dependent manner, and (iii) ICVs transport vRNP to the plasma membrane. How the vRNPs dislocate from ICVs to the inner surface of the PM is not clear.

METTEM allows high signal-to-noise ratio observation of vRNPs (PB2, polymerase subunit) and cellular markers. Using this method the authors identify a possibly more efficient (or alternative) mechanism of vRNP egress than the current model. This may explain the weak effect of cytoskeletal disruption on overall viral growth observed by others. In the current authors' model, vRNPs can directly access the ER surface after nuclear export from the nuclear pores, which may be more efficient than traversing the MTOC via microtubules. As such whether MTOC transition is a prerequisite or an option for vRNPs egress comes into question. The findings are important for the understanding of the IAV lifecycle.

Comments:

(1) In Fig.3D one can see many vRNPs that are not close to ICVs, and associated with filamentous structures. Are these microtubules? A quantification of ICV-associated and non-associated vRNPs is useful to assess the prevalence of the ER-mediated pathway for viral egress, and whether the population is influenced by nocodazole.

(2) Fig.4(B) The authors mention that no ICVs are detected in the Rab11-S25N cells. Likewise. quantification should be done since this is a key observation.

(3) It is possible that the filaments under the PM is a result of ICV fusion with the PM, or transferred via ER-PM membrane contact sites, and these possibilities need to be considered as well as the touch-and-go hypothesis. Have the authors observed fusion events of ICVs with the PM in experiments similar to Fig.6 and Supp Fig.2D? How prevalent is the 'touching'?

(4) The authors state that ICVs are a new type of Rab11-positive organelle. This is intriguing although in order to fully support their hypothesis, ICVs should be rigorously characterised using markers for known organelles that reside in the ER/Golgi region (not only PDI) using IFA, EM, or biochemistry etc.

- It may be worth purifying the ICVs in infected cells by analytical ultracentrifugation and

biochemically analyse them.

(5) The spiked filaments are a major feature of the ICVs. The literature on membrane trafficking may provide useful clues to what they could be (e.g. SNAREs, tethering filaments, etc.) and should be taken into account in conjunction with points 3 and 4 above.

(6) What are the roles of microtubules and actin?

Based on work by others (Lakdawala et al., 2014 PLoS Path needs to be cited), it is important to examine the impact of cytoskeleton destabilisation on ICVs.

(i) Microtubules are known to regulate kinesin-mediated ER sliding, and could play a role in the full emanation and extension of the ER.

(ii) In endosome recycling, actin plays a role in sequence-dependent recycling mediated by distinct tubular subdomains that are dependent on actin polymerisation.

It is certainly extremely interesting to know if the function of microtubules/actin on IAV egress and budding can be explained in light of the authors' present findings.

- Is microtubule/actin required for ICV formation, ER swelling?

Minor comments:

Overall, indicate the used MOI and h.p.i. where they are missing in the legends.

Fig.3 the time point of the images are missing from the text.

Fig.3 Include METTEM in the heading of the legends to distinguish from previous methods.

Supp. Fig5A. the arrows/arrowhead seems to have shifted to the right.

Line107: disulphide-isomerase (PDI) appears for the first time here. Please spell out.

Reviewer #2 (Remarks to the Author):

The RNPs of Influenza virus are composed of the nucleoprotein and the three polymerase subunits PB1, PB2 und PA which are bound to the RNA-genome. Assembly of RNPs occurs in the nucleus and they must then traffic to the plasma membrane, the viral assembly and budding site. It is currently believed that transport occurs via recycling endosomes in a rab11 dependent manner.

Using electron and fluorescence microscopy of virus-infected cells the authors report that virus infection induces remodelling of the ER and the formation of a new organelle, called irregular coated vesicles (ICV). Components of vRNPs and rab11 are localized to the modified ER and to ICVs. Using a dominant negative mutant of rab11 the authors propose that rab11 function is required for ICV formation, that ICVs bud from the ER and touch the plasma membrane. The major claim of the paper is that ICVs (and not recycling endosomes) are the transport vehicle for vRNPs.

This is a novel and interesting finding, but from my point of view most claims are too speculative and not supported by sufficient experimental data.

Origin and nature of ICVs:

One major conclusion is that IVCs are derived from ER membranes, but evidence for that claim is lacking. The authors even show that the luminal ER-marker PDI does not localize to ICVs. Other markers should be tested to prove the ER-origin of the ICVs.

Thus, the claim is mainly based on the observation that ICVs bud from the ER (Fig. 6B). I cannot recognize this process (or the ER itself) on the image presented (arrows). Can the authors exclude this is just close contact instead of budding?

In addition, the criteria for the identification of ICVs are rather vague, translucent lumen and irregular coat. How can the authors exclude that these structures are recycling endosomes (RE)? How do recycling endosomes look like on their TEM images? Can these two vesicle types (ICVs vs. RE) be clearly distinguished as separate classes – for example by immunogold labelling of distinct markers? Only then it seems justified to postulate ICVs as a new vesicle type or new organelle.

The authors report that HA and NA of Influenza virus are not required for remodelling of the ER and formation of ICVs. In the discussion, they speculate that there is an interplay between trafficking of M2 and vRNPs (since both depend on Rab11) and that M2 plays a role in ER modification and ICV formation. This speculation should be investigated using cells expressing M2.

Modified ER:

By describing the kinetics of ER remodelling during infection, the authors give the impression that the ER close to the nucleus expands more and more, finally forming a network between nuclear envelope and plasma membrane (Fig. 2; p.7). However, the ER covers most of the cytoplasm also in non-infected cells and it is already known that it contacts both the plasma membrane and the nuclear envelope. If there were indeed more contact sites established between ER and PM in infected cells, this needs to be shown by quantitative data.

Furthermore, 3D reconstruction of the ER in non-infected cells would be desirable in addition to 3D reconstruction of the ER in infected cells (Fig. 6A) to visualize infection-induced modifications of the ER more clearly.

Methodological evaluation and quantification of the metal-tagging approach:

Imaging of PB2 by metal-tagging (Fig. 3A) reveals PB2 signal along the nuclear membrane and along ER, however very little signal can be seen on one ICV and no signal on another one (3A, arrows); further no signal can be detected on numerous ICVs in Fig 3B, C, D. If the vesicle coat was indeed caused by vRNPs attached to the vesicle as proposed in the paper, shouldn't there be a clear PB2 signal on all ICVs and also inside released virions (3A)? How efficient is this metal labelling method?

Metal-tagging of Rab11 clearly yields staining along ER structures and diffuse signals around vesicles referred to as ICVs (Fig. 4A). However, the signals are locally restricted. What is the meaning of these results? Does this mean there is no Rab11 on ER regions lacking the staining? Or is the labelling efficiency of the applied method rather poor? The same can be seen for ICVs. ICVs on the left (Fig. 4A) lack any Rab11 labeling. Does this mean they are devoid of Rab11, or is this an artefact?

Dominant-negative inactive Rab11-S25N is reported to be found on small vesicles, but not on the ER (Fig. 4B). Could these translucent 400-nm vesicles be recycling endosomes? Why is Rab11(-S25N) detected only in proximity to vesicles but not closely associated to them? Are Rab11-S25N signals not rather randomly distributed?

Overall, quantification of the data would be helpful to convince the reader of the major claims of the manuscript. It is not obvious whether the very few NP-immunogold signals close to ER membranes and PB2 stainings represent significant, non-random colocalization due to specific association between vRNPs and ER or whether they might be random superpositions /encounters.

Furthermore, signals from PB2 and also NP are not necessarily derived from RNPs. Both viral proteins are synthesized in large amounts on cytosolic ribosomes and are then transported to the

nucleus for replication of the viral genome and RNP assembly. Thus, how can signal from monomeric PB2 and NP be discriminated from signals from RNPs?

In sum, the major claims about the origin of ICVs need to be substantiated by more experiments and also by quantification of the data.

Reviewer #3 (Remarks to the Author):

The manuscript by Martin et al, described a new and very interesting finding that the ER is remodeled by RNPs and they report on the existence of small irregularly coated vesicles that form in a manner dependent on Rab11. The finding is very interesting and the work is technologically at a very high level. These insights are novel and unprecedented and will be of interest of scientists from membrane biology as well as virology and thus think that the manuscript is suitable for Nature Communications. Nevertheless, there are few points that have to be addressed before publication.

1- Supplementary Figure 5 is not really convincing: in panel 5A, I don't see PDI-staining gold particles. I don't see a clear difference between ICVs and ER cisternae. Is it possible to add a better image? In panel B, the small and big gold particles seem to always coincide. It is hard for me to explain why this is the case. There is also no quantification of the EM images. I am aware that this is tricky-. EM might not be a quantitative method, but it is important to give the readers an impression of how many images were examined and how often did the authors observe these phenomena.

2- I am aware that the authors discuss possible reasons why fluorescence microscopy studies have missed the remodeling of the ER. I am just not convinced that this is a valid argument. Often the answer to such questions is simply because nobody looked carefully enough. I think that it is important to show the presence of Rab11 (endogenous or overexpressed) on the remodeled ER. In particular, the findings in Figure 4 need to be shown by IF, or by live imaging. I understand the argument on the dynamic range of IF images. However, if Rab11 is present at low concentrations at the remodeled ER, we would expect to see staining pattern that is reminiscent of the ER and we might see Rab11 tubules following the same pattern as ER tubules.

3- Does Rab11 bind to the ER only in infected cells? Is this something triggered by the IAV?

4- Although the phenomenon is very interesting, the story suffers from a lack of mechanism. Ideally, we would like to know exactly how Rab11 drives the formation of the ICVs. However, even known what is not involved, would also be an insight and improve the depth of the story. At the moment, there is no attempt to understand how Rab11 promotes vesicles budding, and what factors mediate (or do not mediate) this effect. Is Rab11 bending membranes and directly supports budding?

5- How would Rab11 be recruited to ER membranes? Is Rab11 recruited to the ER in all areas of the cell, or only in the ER regions that are closely located next to the ERC? It might be that organelle contact sites play a role, which is something that could be investigated in future studies. This type of information on the Rab11 distribution would require fluorescence microscopy, which is another reason why I have asked for it.

Altogether, I think this is a really interesting and well performed work, but that some work is needed to increase the level of confidence and to provide more insights to readers.

Reviewer #4 (Remarks to the Author):

The authors claim that a new organelle called ICVs carry vRNPs, are Rab11 (positive in line with existing literature) and are the means by which vRNP reach the plasma membrane. Indeed, the authors show that upon infection, the cell displays an increased number of ICVS mostly round the MTOC but also further away. The spikes displayed by ICVs are the same seen at the plasma membrane.

The authors also document a remodeling of the ER that extends to the PM as well as ribosome clustering.

The ICVs are found in close proximity to the ER that is also claimed to be positive for Rab11. The authors are just short of claiming that ICVs bud from the ER (although their model clearly claims it). At least they claim that the ER is the primary vector used by ICVs to reach the plasma membrane.

Although potentially interesting, the paper falls short on many aspects, the largest being the lack of demonstration that ICVs bud from the ER (point 6 below). As such it is not suitable for publication at Nature Comm.

Specific comments

1) It is not written in an optimal fashion. The emotions (massive, huge) are in the way of a proper and calmer description of the cell morphology upon virus infection. In figure 1, what we see an accumulation of 100nm coated structures neat the MTOC as well as further away near the plasma membrane.

The MTOC region is different than that on non-infected cells but this is due to the presence of ICVs. The ER remodeling or changes are not well illustrated in this figure (1A-D).

The ER remodeling is way better seen in Figure 2, with more three way junctions (what the authors call tubulation, but this is not demonstrated).

The first two parts of the results need to be re-written to convey these two messages one by one.

2) The ER remodeling is accompanied by a striking change in the ribosome behavior as upon infection, they now form these clusters (Figure 1E). This is as important as the ER remodeling and in my opinion could be linked (see below). This should be presented together with Figure 2 as it is related to the ER.

3) There is no attempt of quantification of any of these changes. EM without quantitative data remains qualitative!

At least the authors should express the percentage of cells exhibiting one phenotype (accumulation of ICVS near MTOC), the ER remodeling and the ribosome clusters? Is there a correlation between these three phenotypes? Are the three phenotypes always present in the same cells etc etc.

Figure 3 and 4 should also be quantified. How many ICVs are decorated compared to total etc. etc. These figures should also be better described in the text.

4) Except for the beautiful IEM, the rest EM is not of very good quality. Especially the micrographs of the METEM are not publishable as they are out of focus (3C, 3D) and the ultrastructure is hardly visible. 3B is in focus but the resolution is poor. This is difficult as the paper technique is mostly EM and the pictures should be stunning.

5) in 3A (also not of great quality), the authors claim that the metal deposition is on the ER. I am not convinced. First, it looks like only a small portion of the ER is decorated. Second, as the picture is enlarged to the maximum allowed by the resolution, the metal is near the ER, not associated to

the membrane. Third, it could just as well be a tubular endosome that are normally positive for Rab11 (see 4A). Both patterns are very similar and Rab11 is hardly on the ER (by IEM).

6) I have no doubt that NP associated to ICVs (that are also positive for Rab11) and that these structures are involved in the virus biology. This is well documented.

What I am not convinced of at all is the connection to the ER.

-ICVS are close to the ER but not in contact as claimed by the authors. These days, organelle contact is defined as less than 30nm. The pictures provided show a close proximity but no contact as such. Even the 3D tomography does not show either contact or budding (see below).

-I also do not see any convincing budding profiles for ICV from the ER. The picture in 6B could suggest it, but the tubular profile is not ER as it is covered by spikes.

-The ER labeling for NP is marginal. Granted, NP labeling is close to ER but not on it (IEM plus my comment above about METEM).

-No PDI is found in the ICVs.

At best, what the authors show is that the ICVs are close to the ER that tend to pervade the entire cell. Whether the ER is involved/required in the budding of ICVs remain to be demonstrated.

7) There is also no functional data. Can the link of NP (vRNP) to ER be compromised?

The experiment with the Rab11 mutant is interesting but it addresses the role of Rab11 in the recruitment of NP to spiky ICVS, not the role of the ER in their formation.

Therefore, the first conclusion of this article is that virus RNPs (marked by NP) are associated to 100-200nm Rab11 positive structures that could (or not) be small endosomes. This has not been addressed in any way. No endosomal markers in addition to Rab11 has been used to attempt to identify ICVs.

In addition, there is no attempt to characterize the spiky coat. Can the authors at least rule out the involvement of clathrin. I am not saying that the entire coat of clathrin but it would be present there!

The second conclusion of the paper is that the ER is extensively remodeled and that the ribosomes clusters. Whether it has to do with the ICVs formation remains to be demonstrated. It is also (likely) possible that the ribosome clustering leads to the ER tubulation. The relationship between ribosome presence on the ER and its sheet/tubule morphology has been addressed in an old paper from the Jokitalo's group.

IS there a relationship between ribosome clustering and infection, formation of ICVs. Could the experiment in Figure 2 (kinetics) be quantified to inform on the link between these phenotypes?

Minor

-Add indication on the figures, gold size corresponding to what, time of infection, color coding for the IF etc. It will make the reading much easier.

-Rab11 labeling seems to decrease at the peripheral region

-deconvolute 2E to appreciate the strong remodeling of the ER upon infection

Response to reviewers of Manuscript NCOMMS-17-01013

« A new class of Rab11-dependent vesicles transports influenza virus genome from a modified endoplasmic reticulum to the plasma membrane »

We would like to thank the reviewers for their comments and suggestions. In the revised version of the manuscript, the text and figures have been modified in order to address the reviewers' concerns. A revised version of the manuscript with highlighted changes is provided.

Below is a point-by-point response to each of the reviewers' comments. The page and line numbers refer to the revised version of the manuscript.

Reviewer #1 (Remarks to the Author):

A new class of Rab11-dependent vesicles transports influenza virus genome from a modified endoplasmic reticulum to the plasma membrane
de Castro Martin et al. Nature Communications

Influenza A virus (IAV) vRNPs require Rab11, a recycling endosomal GTPase, for egress toward the plasma membrane. After replication in the nucleus, progeny vRNPs are nuclear exported by a CRM1-dependent pathway dependent on matrix protein M1 and nuclear export protein (NEP), and facilitated by cellular factors such as YB-1. After initial concentration at the MTOC, it has been suggested that vRNP egress occurs by piggy-backing on Rab11-positive recycling endosomes that traffic outward to the cell periphery along microtubules. Rab11-depletion or dominant-negative expression reduces the release of infectious progeny virions (Eisfeld et al., 2015; Lakdawala et al., 2016).

In this study, de Castro-Martin et al. show using conventional EM, cryo EM and metal-tagging TEM (METTEM) that progeny vRNPs are found on the surface of irregularly coated vesicles (ICVs), endomembrane organelles that bud from the ER and touch the plasma membrane (PM). Infected cells produce new organelles consisting of ICVs on which progeny vRNPs are present but protein disulphide-isomerase (PDI) is not. They propose an alternative model of vRNP egress: (i) vRNPs locate to the ER following export, (ii) ICVs displaying Rab11/vRNPs bud from the ER in a Rab11-dependent manner, and (iii) ICVs transport vRNP to the plasma membrane. How the vRNPs dislocate from ICVs to the inner surface of the PM is not clear.

METTEM allows high signal-to-noise ratio observation of vRNPs (PB2, polymerase subunit) and cellular markers. Using this method the authors identify a possibly more efficient (or alternative) mechanism of vRNP egress than the current model. This may explain the weak effect of cytoskeletal disruption on overall viral growth observed by others. In the current authors' model, vRNPs can directly access the ER surface after nuclear export from the nuclear pores, which may be more efficient than traversing the MTOC via microtubules. As such whether MTOC transition is a prerequisite or an option for vRNPs egress comes into question. The findings are important for the understanding of the IAV lifecycle.

Comments:

(1) In Fig.3D one can see many vRNPs that are not close to ICVs, and associated with filamentous structures. Are these microtubules? A quantification of ICV-associated and non-associated vRNPs is useful to assess the prevalence of the ER-mediated pathway for viral egress, and whether the population is influenced by nocodazole.

In the revised version of the manuscript (Supp. Table 2, page 9 lines 18-21), we provide a quantification of vRNPs observed upon SE-METTEM in association with the nuclear envelope / the ER

/ filaments / ICVs / in the cytosol, for n=29 cells. The data show that the ICV-associated and ER-associated vRNPs are significantly more abundant than filament-associated vRNPs ($p < 0.0001$, Wilcoxon matched-pairs signed-rank test). These data strengthen the notion that the ER-mediated pathway for vRNP egress is predominant. They are in agreement with previous reports by us (Avilov et al, J Virol 2012) and by others (e.g. Amorim et al, J Virol 2011; Kumakura et al, Virology 2015) showing that microtubule disruption or depolymerisation of actin filaments have a mild effect on the production of infectious virions. Therefore we considered unlikely that cytoskeleton destabilisation would have a strong impact on the population of ICVs.

(2) Fig.4(B) The authors mention that no ICVs are detected in the Rab11-S25N cells. Likewise, quantification should be done since this is a key observation.

In the revised version of the manuscript (Supp. Table 3), we provide a quantification of Rab11 signal found in association with the nuclear envelope / the ER / ICVs / smooth vesicles, on Rab11-S25N cells and wt-Rab11 cells (n=25 for each type of cells). No ICVs were detected in Rab11-S25N cells whereas as total of 143 ICVs associated with a Rab11 signal were detected in wt-Rab11 cells. These observations are mentioned page 11 lines 13-17.

(3) It is possible that the filaments under the PM is a result of ICV fusion with the PM, or transferred via ER-PM membrane contact sites, and these possibilities need to be considered as well as the touch-and-go hypothesis. Have the authors observed fusion events of ICVs with the PM in experiments similar to Fig.6 and Supp. Fig.2D? How prevalent is the 'touching'?

In the revised version of the manuscript, we provide a quantification of the number of ICVs that are close (< 200 nm) or distant (> 200 nm) to the PM, making a distinction between ICVs with a dense irregular coat (observed as single vesicles or as pairs), and ICVs with a sparse irregular coat (always observed as single vesicles), on n=25 cells. The data, as shown in Supp. Table 4 and stated page 12 lines 8-14, indicate that ICVs with a sparse irregular coat are mainly observed close to the PM, and are in favor of our proposed hypothesis of a touch-and-go mechanism. We have not observed any ICV fusion with the PM (stated page 12 lines 14-15). However, we cannot exclude the possibility that the filaments can also be transferred to the PM via ER-PM membrane contact sites, as stated page 12 lines 18-20.

(4) The authors state that ICVs are a new type of Rab11-positive organelle. This is intriguing although in order to fully support their hypothesis, ICVs should be rigorously characterised using markers for known organelles that reside in the ER/Golgi region (not only PDI) using IFA, EM, or biochemistry etc. It may be worth purifying the ICVs in infected cells by analytical ultracentrifugation and biochemically analyse them.

In the revised version of the manuscript, we present additional data that unambiguously demonstrate that ICVs are a new type of Rab11-positive organelle distinct from recycling endosomes
1) Double-labeling by immuno-EM was performed for Rab11 and the transferrin receptor (TfR), a marker of recycling endosomes. Rab11 and TfR frequently co-localized to small vesicles with a morphology characteristic of recycling endosomes, notably an electron-dense lumen. In contrast, the vast majority of ICVs showed Rab11 signal but were devoid of TfR signal, which together with their electron-lucent lumen clearly differentiated them from recycling endosomes (Supp. Fig. 14 and page 11 lines 5-11 of the revised manuscript).

2) Upon subcellular fractionation by sucrose gradient ultracentrifugation, the peak concentration of the Rab11 and TfR markers of recycling endosomes coincided in the 40-30% fraction from mock-infected cells whereas in IAV-infected cells Rab11 and TfR concentration peaked in the 60-40% and 40-30% fractions, respectively. This difference, taken together with the fact that ICVs were detected in the 60-40% fraction but not in the 40-30% fraction of infected cells upon negative staining, and the fact that Rab11 co-fractionated with NP and PB2 in the 60-40% fraction, provides additional evidence that ICVs are distinct from recycling endosomes (Fig. 9 and page 12 lines 22-27 and page 13 lines 1-10 of the revised manuscript).

3) Upon subcellular fractionation by sucrose gradient ultracentrifugation, the peak concentration of calreticulin coincided with the peak concentration of Rab11 in IAV-infected cells (in the 60-40% fraction) whereas the calreticulin peak was detected mostly in the 40-30% fraction from mock-infected cells (Fig. 9 and page 13 lines 12-18 of the revised manuscript).

(5) The spiked filaments are a major feature of the ICVs. The literature on membrane trafficking may provide useful clues to what they could be (e.g. SNAREs, tethering filaments, etc.) and should be taken into account in conjunction with points 3 and 4 above.

In the revised version of the manuscript we present electron microscopy images that highlight the morphological difference between ICVs and clathrin-coated vesicles or caveolae (Supp. Fig 2).

In addition, we have measured the length of ICV filaments (n=137 filaments from 16 ICVs). They range in length from 26.6 to 112.5 nm, which is consistent with the reported length of the vRNPs (Supp. Fig. 8, page 8 lines 23-25 of the revised version).

We have also performed double labeling for PB2 and NP, and observed that both proteins can be detected at the membrane of ICVs (Supp. Fig. 11, page 9 lines 24-27 and page 10 lines 1-4 of the revised version).

Finally, upon negative staining of a subcellular fraction enriched in ICVs, the irregular coat shows the same morphology as isolated vRNPs (Sugita et al, J Virol 2013) (Fig. 9B and page 13 lines 1-3).

Taken altogether, these observations clearly demonstrate that the spiked filaments at the surface of ICVs are vRNPs.

(6) What are the roles of microtubules and actin?

Based on work by others (Lakdawala et al., 2014 PLoS Path needs to be cited), it is important to examine the impact of cytoskeleton destabilisation on ICVs.

The paper from Lakdawala et al. (2014 PLoS Path) was cited as reference #12 in the first version of our manuscript (reference #12 in the revised version as well).

(i) Microtubules are known to regulate kinesin-mediated ER sliding, and could play a role in the full emanation and extension of the ER.

(ii) In endosome recycling, actin plays a role in sequence-dependent recycling mediated by distinct tubular subdomains that are dependent on actin polymerisation.

It is certainly extremely interesting to know if the function of microtubules/actin on IAV egress and budding can be explained in light of the authors' present findings.

- Is microtubule/actin required for ICV formation, ER swelling?

We agree with the reviewer that microtubules and/or actin could play a role in ER-reshaping and ICV formation. However, given (i) the fact that we and others have found that microtubule disruption or depolymerisation of actin filaments have only a mild effect on the production of infectious virions (e.g. Avilov et al, J Virol 2012; Amorim et al, J Virol 2011; Kumakura et al, Virology 2015), and (ii) our observation that ICVs are not predominantly associated with cytoskeletal filaments (Supp. Table 2, page 9 lines 18-21), we considered it unlikely that cytoskeleton destabilisation would have a strong impact on the formation of ICVs.

Minor comments:

Overall, indicate the used MOI and h.p.i. where they are missing in the legends.

The used MOI is 5 pfu/cell throughout the study, as indicated in the Materials and Methods section (page 19 lines 17-18). The h.p.i. have been systematically indicated in the figure legends.

Fig.3 the time point of the images are missing from the text.

The time points of Fig. 3 (Fig. 4 in the revised version) (8 hpi) has been indicated in the text (page 9 line 13).

Fig.3 Include METTEM in the heading of the legends to distinguish from previous methods.
The heading of the legends have been modified as requested by the reviewer.

Supp. Fig5A. the arrows/arrowhead seems to have shifted to the right.
The Supp. Fig5 (Supp. Fig 12 in the revised version) has been improved.

Line107: disulphide-isomerase (PDI) appears for the first time here. Please spell out.
Protein disulphide-isomerase has been spelled out (page 10 line 5).

Reviewer #2 (Remarks to the Author):

The RNPs of Influenza virus are composed of the nucleoprotein and the three polymerase subunits PB1, PB2 und PA which are bound to the RNA-genome. Assembly of RNPs occurs in the nucleus and they must then traffic to the plasma membrane, the viral assembly and budding site. It is currently believed that transport occurs via recycling endosomes in a rab11 dependent manner. Using electron and fluorescence microscopy of virus-infected cells the authors report that virus infection induces remodelling of the ER and the formation of a new organelle, called irregular coated vesicles (ICV). Components of vRNPs and rab11 are localized to the modified ER and to ICVs. Using a dominant negative mutant of rab11 the authors propose that rab11 function is required for ICV formation, that ICVs bud from the ER and touch the plasma membrane. The major claim of the paper is that ICVs (and not recycling endosomes) are the transport vehicle for vRNPs.

This is a novel and interesting finding, but from my point of view most claims are too speculative and not supported by sufficient experimental data.

(1) Origin and nature of ICVs:

One major conclusion is that IVCs are derived from ER membranes, but evidence for that claim is lacking. The authors even show that the luminal ER-marker PDI does not localize to ICVs. Other markers should be tested to prove the ER-origin of the ICVs.

Thus, the claim is mainly based on the observation that ICVs bud from the ER (Fig. 6B). I cannot recognize this process (or the ER itself) on the image presented (arrows). Can the authors exclude this is just close contact instead of budding?

In addition, the criteria for the identification of ICVs are rather vague, translucent lumen and irregular coat. How can the authors exclude that these structures are recycling endosomes (RE)? How do recycling endosomes look like on their TEM images? Can these two vesicle types (ICVs vs. RE) be clearly distinguished as separate classes – for example by immunogold labelling of distinct markers? Only then it seems justified to postulate ICVs as a new vesicle type or new organelle.

In the revised version of the manuscript, we provide zooms of budding zones and strong evidence for the continuity between the ER membrane and the ICVs, in two Supplementary figures (Supp. Fig. 15 and 16). The text (page 12 lines 4-7) has been modified accordingly.

In addition, we present new data that unambiguously demonstrate that ICVs are a new type of Rab11-positive organelle distinct from recycling endosomes

1) Double-labeling by immuno-EM was performed for Rab11 and the transferrin receptor (TfR), a marker of recycling endosomes. Rab11 and TfR frequently co-localized to small vesicles with a morphology characteristic of recycling endosomes, notably an electron-dense lumen. In contrast, the vast majority of ICVs showed Rab11 signal but were devoid of TfR signal, which together with their electron-lucent lumen clearly differentiated them from recycling endosomes (Supp. Fig. 14 and page 11 lines 5-11 of the revised manuscript).

2) Upon subcellular fractionation by sucrose gradient ultracentrifugation, the peak concentration of

the Rab11 and TfR markers of recycling endosomes coincided in the 40-30% fraction from mock-infected cells whereas in IAV-infected cells Rab11 and TfR concentration peaked in the 60-40% and 40-30% fractions, respectively. This difference, taken together with the fact that ICVs were detected in the 60-40% fraction but not in the 40-30% fraction of infected cells upon negative staining, and the fact that Rab11 co-fractionated with NP and PB2 in the 60-40% fraction, provides additional evidence that ICVs are distinct from recycling endosomes (Fig. 9 and page 12 lines 22-27 and page 13 lines 1-10 of the revised manuscript).

(2) The authors report that HA and NA of Influenza virus are not required for remodelling of the ER and formation of ICVs. In the discussion, they speculate that there is an interplay between trafficking of M2 and vRNPs (since both depend on Rab11) and that M2 plays a role in ER modification and ICV formation. This speculation should be investigated using cells expressing M2.

Our hypothesis was investigated using cells treated prior to infection with siRNA targeting M2. Upon electron microscopy or indirect immunofluorescence staining of Rab11 and NP, little difference was observed between M2-depleted cells and control cells, which argues against a critical role for M2 in ER modification and ICV formation. These data are presented in Supp. Fig. 6 and page 8 lines 7-12 of the revised version of the manuscript, and discussed page 17 lines 3-7.

(3) Modified ER:

By describing the kinetics of ER remodelling during infection, the authors give the impression that the ER close to the nucleus expands more and more, finally forming a network between nuclear envelope and plasma membrane (Fig. 2; p.7). However, the ER covers most of the cytoplasm also in non-infected cells and it is already known that it contacts both the plasma membrane and the nuclear envelope. If there were indeed more contact sites established between ER and PM in infected cells, this needs to be shown by quantitative data.

Furthermore, 3D reconstruction of the ER in non-infected cells would be desirable in addition to 3D reconstruction of the ER in infected cells (Fig. 6A) to visualize infection-induced modifications of the ER more clearly.

In the revised version of the manuscript, we present serial sections that clearly show the difference in ER organisation between mock- and IAV-infected cells (Fig. 2, page 7 lines 2-5).

We also provide images to show that the contact sites established between ER elements and the plasma membrane differ between mock- and IAV-infected cells (Supp. Fig. 5, page 7 lines 26-27 and page 8 lines 1-6).

(4) Methodological evaluation and quantification of the metal-tagging approach:

Imaging of PB2 by metal-tagging (Fig. 3A) reveals PB2 signal along the nuclear membrane and along ER, however very little signal can be seen on one ICV and no signal on another one (3A, arrows); further no signal can be detected on numerous ICVs in Fig 3B, C, D. If the vesicle coat was indeed caused by vRNPs attached to the vesicle as proposed in the paper, shouldn't there be a clear PB2 signal on all ICVs and also inside released virions (3A)? How efficient is this metal labelling method?

Metal-tagging of Rab11 clearly yields staining along ER structures and diffuse signals around vesicles referred to as ICVs (Fig. 4A). However, the signals are locally restricted. What is the meaning of these results? Does this mean there is no Rab11 on ER regions lacking the staining? Or is the labelling efficiency of the applied method rather poor? The same can be seen for ICVs. ICVs on the left (Fig. 4A) lack any Rab11 labeling. Does this mean they are devoid of Rab11, or is this an artefact?

METTEM is a labeling method of high sensitivity. Our previous works demonstrated that labeling efficiency is superior to that provided by antibodies by at least two orders of magnitude (Risco et al., Structure 2012; Fernández de Castro et al., Methods Cell Biol 2014). This efficiency is due to the fact that whole cells are labeled in vivo and protein detection is not restricted to the surface of ultrathin

sections (Fernandez de Castro et al., J. Cell Sci. 2017). Here we have further improved the sensitivity by developing the silver enhancement step. However our method is not expected to provide absolute sensitivity for the following reasons:

i) the efficiency of the silver enhancement reaction might depend on the number of gold atoms bound by the MT-tags, and it is likely that not all MT tags bind to enough gold atoms to allow the growth of gold-silver nanoclusters

ii) the Rab11 protein is known to accumulate in membranous micro-domains (e.g. Sönnichsen et al, 2000), and may not be homogeneously distributed, which could account for the fact that not all ER regions and not all ICVs are equally labeled with Rab11-MT.

iii) when fused to PB2 in the context of the vRNPs, the accessibility of the MT-tag to the products of the enhancement reaction, and therefore the growth of gold-silver nanoclusters, might be limited. This hypothesis is supported by the fact that some virions are not labeled with an anti-PB2 antibody (Supp. Fig. 11). In addition, we have observed that many ICVs have few filaments (Supp. Table 4), which could also result in a weak labeling.

Dominant-negative inactive Rab11-S25N is reported to be found on small vesicles, but not on the ER (Fig. 4B). Could these translucent 400-nm vesicles be recycling endosomes? Why is Rab11(-S25N) detected only in proximity to vesicles but not closely associated to them? Are Rab11-S25N signals not rather randomly distributed?

A zoom has been included in Fig. 4B (renumbered Fig. 6B in the revised version of the manuscript) to show that Rab11-S25N labeling is indeed on small vesicles. This observation is consistent with the results from immunogold labeling (Fig 7C-7D) and with previous observations by Ullrich et al. (J Cell Biol 1993) that Rab11-S25N can be found on vesicles and tubules. As Ullrich et al. also demonstrated that Rab11-S25N reduces the recycling, we do not expect the small Rab11-S25 positive vesicles to be recycling endosomes.

Overall, quantification of the data would be helpful to convince the reader of the major claims of the manuscript. It is not obvious whether the very few NP-immunogold signals close to ER membranes and PB2 stainings represent significant, non-random colocalization due to specific association between vRNPs and ER or whether they might be random superpositions /encounters.

In the revised version of the manuscript, we provide a quantification of SE-METTEM PB2 staining found in association with the nuclear envelope / the ER / filaments / ICVs / in the cytosol, for n=29 cells (Supp. Table 2, page 9 lines 18-21). The data show that the PB2 signal is significantly more frequently associated to the nuclear envelope, ER and ICVs compared to the filaments or cytosol ($p < 0.0001$, Wilcoxon matched-pairs signed-rank test). These data argue against a random colocalisation of PB2 and ER membranes, and strengthen the notion that the ER-mediated pathway for vRNP egress is predominant.

Furthermore, signals from PB2 and also NP are not necessarily derived from RNPs. Both viral proteins are synthesized in large amounts on cytosolic ribosomes and are then transported to the nucleus for replication of the viral genome and RNP assembly. Thus, how can signal from monomeric PB2 and NP be discriminated from signals from RNPs?

We agree with the reviewer that our staining methods for PB2 and NP cannot discriminate monomeric proteins from vRNPs. However, published observations on leptomycin-treated cells indicate that at late time points (i.e. > 6 hours post-infection), the PB2 and NP signals detected in the cytoplasm of IAV-infected cells correspond mostly to neo-synthesized vRNPs (e.g. Avilov et al., J Virol 2012). Furthermore, we performed PB2 + NP double-labeling by immuno-EM: ICVs showed signal for both NP and PB2, strengthening our conclusion that they are coated with vRNPs (Supp. Fig. 11, page 9 lines 24-27 and page 10 lines 1-4 of the revised version).

In addition, upon negative staining of a subcellular fraction enriched in ICVs, the irregular coat of the ICVs shows the same morphology as isolated vRNPs (Sugita et al, J Virol 2013) (Fig. 9B and page 13 lines 1-3), unequivocally demonstrating the presence of vRNPs on ICVs.

In sum, the major claims about the origin of ICVs need to be substantiated by more experiments and also by quantification of the data.

Taken altogether, the additional experimental data and quantification data presented in the revised version of the manuscript (please see also the response to other reviewers' comments) provide very strong evidence for the fact that ICVs derive from the ER and are distinct from recycling endosomes.

Reviewer #3 (Remarks to the Author):

The manuscript by Martin et al, described a new and very interesting finding that the ER is remodeled by RNPs and they report on the existence of small irregularly coated vesicles that form in a manner dependent on Rab11. The finding is very interesting and the work is technologically at a very high level. These insights are novel and unprecedented and will be of interest of scientists from membrane biology as well as virology and thus think that the manuscript is suitable for Nature Communications. Nevertheless, there are few points that have to be addressed before publication.

1- Supplementary Figure 5 is not really convincing: in panel 5A, I don't see PDI-staining gold particles. I don't see a clear difference between ICVs and ER cisternae. Is it possible to add a better image? In panel B, the small and big gold particles seem to always coincide. It is hard for me to explain why this is the case.

The Supp. Fig. 5 has been improved along with the suggestions of the reviewer (Supp. Fig. 12 in the revised version).

There is also no quantification of the EM images. I am aware that this is tricky-. EM might not be a quantitative method, but it is important to give the readers an impression of how many images were examined and how often did the authors observe these phenomena.

In the revised version of the manuscript, quantification data for the EM images is provided (Supp. Tables 1-4, page 7 lines 7-10, page 9 lines 18-21, page 11 lines 15-17, page 12 lines 8-15).

2- I am aware that the authors discuss possible reasons why fluorescence microscopy studies have missed the remodeling of the ER. I am just not convinced that this is a valid argument. Often the answer to such questions is simply because nobody looked carefully enough. I think that it is important to show the presence of Rab11 (endogenous or overexpressed) on the remodeled ER. In particular, the findings in Figure 4 need to be shown by IF, or by live imaging. I understand the argument on the dynamic range of IF images. However, if Rab11 is present at low concentrations at the remodeled ER, we would expect to see staining pattern that is reminiscent of the ER and we might see Rab11 tubules following the same pattern as ER tubules.

In the revised version of the manuscript we provide new immunofluorescence and confocal microscopy on mock- and IAV-infected cells, stained for Rab11 and PDI (Fig.5, page 10 lines 18-23). Infected cells show several Rab11-PDI colocalization spots, suggesting a proximity between Rab11-positive vesicles and the tubular ER elements. We did not observe Rab11 tubular structures, which we believe is related to a low density of Rab11 in the ER.

3- Does Rab11 bind to the ER only in infected cells? Is this something triggered by the IAV? METTEM images provided in Supp. Fig. 6 (re-numbered Supp. Fig. 13 in the revised version) show that in mock-infected cells, Rab11 is detected in vesicles and sporadically also on ER cisternae. In contrast, in IAV-infected cells, a strong Rab11 signal can be detected on large portions of the remodeled ER (Fig. 4A, re-numbered Fig. 6A in the revised version), suggesting that that IAV infection triggers the relocation and/or redistribution of Rab11 proteins in the ER. These observations are mentioned page 10 lines 24-27 and discussed page 16 lines 6-12 of the revised version of the manuscript.

4- Although the phenomenon is very interesting, the story suffers from a lack of mechanism. Ideally, we would like to know exactly how Rab11 drives the formation of the ICVs. However, even known what is not involved, would also be an insight and improve the depth of the story. At the moment, there is no attempt to understand how Rab11 promotes vesicles budding, and what factors mediate (or do not mediate) this effect. Is Rab11 bending membranes and directly supports budding?

By documenting the endomembrane system in cells that express the inactive, dominant-negative mutant form Rab11-S25N, the initial version of our manuscript provided some substantial mechanistic information. Indeed, our observations demonstrate that i) active Rab11 GTPases are not necessary for IAV-induced ER remodeling, ii) only active Rab11 GTPases are accumulating in some regions of the ER in infected cells, and iii) active Rab11 GTPases are necessary for ICV formation (Fig. 4 and Fig.5 of the initial version, Fig. 6, Fig. 7, and Supp. Table 3 of the revised version). These observations are mentioned page 11 lines 12-23 and page 15 lines 25-27 of the revised version of the manuscript.

In addition, we demonstrate that the three viral membrane proteins (HA, NA and M2) are not required for ICV formation (Supp. Fig 6 and Supp. Fig. 7, page 8 lines 7-18 and page 16 lines 22-27, page 17 lines 1-7 of the revised version of the manuscript).

5- How would Rab11 be recruited to ER membranes? Is Rab11 recruited to the ER in all areas of the cell, or only in the ER regions that are closely located next to the ERC? It might be that organelle contact sites play a role, which is something that could be investigated in future studies. This type of information on the Rab11 distribution would require fluorescence microscopy, which is another reason why I have asked for it.

The IF and EM data provided in the revised version of the manuscript (Fig. 5 and Fig. 6, respectively) show that the presence of Rab11 at the ER membrane is not limited to the vicinity of the ERC.

However, as stated page 10 lines 26-27, Rab11 is not homogeneously distributed in all areas of the ER. Organelle contacts sites might play a role in Rab11 recruitment to ER membranes: this hypothesis has been mentioned in the Discussion section (page 16 lines 9-10).

Altogether, I think this is a really interesting and well performed work, but that some work is needed to increase the level of confidence and to provide more insights to readers.

Reviewer #4 (Remarks to the Author):

The authors claim that a new organelle called ICVs carry vRNPs, are Rab11 (positive in line with existing literature) and are the means by which vRNP reach the plasma membrane.

Indeed, the authors show that upon infection, the cell displays an increased number of ICVs mostly round the MTOC but also further away. The spikes displayed by ICVs are the same seen at the plasma membrane.

The authors also document a remodeling of the ER that extends to the PM as well as ribosome clustering.

The ICVs are found in close proximity to the ER that is also claimed to be positive for Rab11

The authors are just short of claiming that ICVs bud from the ER (although their model clearly claims it). At least they claim that the ER is the primary vector used by ICVs to reach the plasma membrane.

Although potentially interesting, the paper falls short on many aspects, the largest being the lack of demonstration that ICVs bud from the ER (point 6 below). As such it is not suitable for publication at Nature Comm.

Specific comments

1) It is not written in an optimal fashion. The emotions (massive, huge) are in the way of a proper and calmer description of the cell morphology upon virus infection. In figure 1, what we see an accumulation of 100nm coated structures neat the MTOC as well as further away near the plasma

membrane. The MTOC region is different than that on non-infected cells but this is due to the presence of ICVs. The ER remodeling or changes are not well illustrated in this figure (1A-D). The ER remodeling is way better seen in Figure 2, with more three way junctions (what the authors call tubulation, but this is not demonstrated).

In the revised version of the manuscript we present serial sections that clearly show the difference in ER organisation between mock- and IAV-infected cells (Fig. 2, page 7 lines 2-5).

We also provide images to show that the contact sites established between ER elements and the plasma membrane differ between mock- and IAV-infected cells (Supp. Fig. 5, page 7 lines 26-27 and page 8 lines 1-6).

The first two parts of the results need to be re-written to convey these two messages one by one. The text has been modified, to take into consideration the comments of Reviewer #4.

2) The ER remodeling is accompanied by a striking change in the ribosome behavior as upon infection, they now form these clusters (Figure 1E). This is as important as the ER remodeling and in my opinion could be linked (see below). This should be presented together with Figure 2 as it is related to the ER.

As suggested by the reviewer, the text has been modified to better underline the change in ribosome distribution and the potential link to ER remodeling (page 7 lines 10-14 of the revised version of the manuscript. Fig. 1E has not been moved to Fig. 2 (renumbered Fig. 3 in the revised version), in order to preserve the large size of the images shown in Fig. 3 (revised numbering). The ER remodeling is also illustrated in details in the Supp. Fig. 3 (Supp. Fig.7 in the initial version of the manuscript).

3) There is no attempt of quantification of any of these changes. EM without quantitative data remains qualitative!

At least the authors should express the percentage of cells exhibiting one phenotype (accumulation of ICVs near MTOC), the ER remodeling and the ribosome clusters? Is there a correlation between these three phenotypes? Are the three phenotypes always present in the same cells etc etc.

The three phenotypes (accumulation of ICVs near MTOC / ER remodeling / ribosome clusters) have been analyzed in n=18 infected cells. All three phenotypes were present in all 18 infected cells, therefore the three phenotypes induced by IAV infection are potentially linked to each other. These observations are mentioned page 7 lines 10-14 and discussed page 14 lines 24-26 of the revised version of the manuscript.

Figure 3 and 4 should also be quantified. How many ICVs are decorated compared to total etc. etc. These figures should also be better described in the text.

The data shown in Fig. 3 and Fig. 4 (Fig. 4 and Fig. 6 in the revised version) have been quantified (Supp. Table 2 and Supp. Table 3 of the revised version).

We provide a quantification of vRNPs observed upon SE-METEM in association with the nuclear envelope / the ER / filaments / ICVs / in the cytosol, for n=29 cells (Supp. Table 2, page 9 lines 18-21). The data show that the ICV-associated and ER-associated vRNPs are significantly more abundant than filament-associated vRNPs ($p < 0.0001$, Wilcoxon matched-pairs signed-rank test). These data strengthen the notion that the ER-mediated pathway for vRNP egress is predominant.

We also provide a quantification of Rab11 signal found in association with the nuclear envelope / the ER / ICVs / small vesicles, on Rab11-S25N cells and wt-Rab11 cells (n=25 for each type of cells) (Supp. Table 3). No ICVs were detected in Rab11-S25N cells whereas a total of 143 ICVs associated with a Rab11 signal were detected in wt-Rab11 cells. These observations are mentioned page 11 lines 15-17.

4) Except for the beautiful IEM, the rest EM is not of very good quality. Especially the micrographs of the METEM are not publishable as they are out of focus (3C, 3D) and the ultrastructure is hardly visible. 3B is in focus but the resolution is poor. This is difficult as the paper technique is mostly EM

and the pictures should be stunning.

METTEM is a labeling method of high sensitivity. Our previous works demonstrated that labeling efficiency is superior to that provided by antibodies by at least two orders of magnitude (Risco et al., Structure 2012; Fernández de Castro et al., Methods Cell Biol 2014). This efficiency is due to the fact that whole cells are labeled in vivo and protein detection is not restricted to the surface of ultrathin sections (Fernandez de Castro et al., J. Cell Sci. 2017). The silver enhancement step was specifically developed for this work. In this case we have been able to obtain a good compromise of labeling sensitivity and ultrastructural definition that provided new information about the association of IAV vRNPs with different intracellular compartments. It is noteworthy that other laboratories using silver enhancement of ultrasmall gold probes during freeze substitution, also showed results with a morphology that is less well preserved compared to samples that are only processed by high pressure freezing and freeze substitution (He et al., 2007; Morphew et al., 2008). How the silver enhancement reaction compromises morphological preservation is so far unknown, but in the future, the method will be developed further to improve contrast in samples incubated with silver.

5) in 3A (also not of great quality), the authors claim that the metal deposition is on the ER. I am not convinced. First, it looks like only a small portion of the ER is decorated. Second, as the picture is enlarged to the maximum allowed by the resolution, the metal is near the ER, not associated to the membrane. Third, it could just as well be a tubular endosome that are normally positive for Rab11 (see 4A). Both patterns are very similar and Rab11 is hardly on the ER (by IEM).

In the revised version of the manuscript, Fig. 3 (renumbered Fig. 4) has been modified to better show the PB2-MT labeling on the ER membranes (zoomed image in panel 4B).

New immunoEM data are provided which clearly highlight differences between recycling endosomes and ICVs. Recycling endosomes show Rab11 and transferrin receptor labeling and an electron-dense lumen. By contrast, ICVs show Rab11 signal but are almost devoid of transferrin receptor signal, and they show an electron-lucent lumen (Supp. Fig. 14, page 11 lines 5-11). New subcellular fractionation data are also provided, which indicate that the subcellular distribution of Rab11 and ER markers differs between mock- and IAV-infected cells (Fig. 9, page 12 lines 22-27 and page 13 lines 1-18). Taken together, these data confirm that the intra-cellular distribution of Rab11 differs between uninfected and IAV-infected cells, and that ICVs are distinct from recycling endosomes.

6) I have no doubt that NP associated to ICVs (that are also positive for Rab11) and that these structures are involved in the virus biology. This is well documented.

What I am not convinced of at all is the connection to the ER.

-ICVs are close to the ER but not in contact as claimed by the authors. These days, organelle contact is defined as less than 30nm. The pictures provided show a close proximity but no contact as such. Even the 3D tomography does not show either contact or budding (see below).

I also do not see any convincing budding profiles for ICV from the ER. The picture in 6B could suggest it, but the tubular profile is not ER as it is covered by spikes.

In Supp. Fig. 15 and 16 of the revised version of the manuscript, we show higher magnification images of selected areas of Fig. 8B and Supp. Fig. 3C (Fig. 6B and Supp. Fig. 7C in the initial version), to provide strong evidence for the continuity between the ER membrane and the ICV (highlighted by a blue line). The tubular structures are associated with ribosomes (yellow arrows), indicating that they correspond to ER cisterns. The fact that these structures are also covered with spike is very consistent with our METTEM detection of vRNPs on ER membranes (Fig. 4A). The text has been modified to take into account the additional information provided (page 12 lines 4-7).

-The ER labeling for NP is marginal. Granted, NP labeling is close to ER but not on it (IEM plus my comment above about METTEM).

The Supp. Fig. 5 (ImmunoEM for NP and PDI, renumbered Supp. Fig. 12 in the revised version) has been improved to better show the NP labeling on ER membranes.

-No PDI is found in the ICVs.

We would not expect PDI to be present in ICVs since PDI has an KDEL ER-localization motif and thus the majority of it resides in the ER. In this line PDI is only exceptionally found on ER exit sites in non-infected cells (Oprins et al., 1993). Since ICVs bud from the ER as shown in the Supp. Fig. 15 and 16 of the revised manuscript, the absence of PDI within their lumen shows that their budding is not a bulk flow process and they do not represent a subdomain of the ER.

At best, what the authors show is that the ICVs are close to the ER that tend to pervade the entire cell. Whether the ER is involved/required in the budding of ICVs remain to be demonstrated.

We believe that the revised version of the manuscript now provides strong evidence that ICVs are budding from the ER (Fig 8B, Supp. Fig. 3C, Supp. Fig. 15 and Supp. Fig. 16) and are distinct from recycling endosomes (Supp. Fig. 14, Fig. 9).

7) There is also no functional data. Can the link of NP (vRNP) to ER be compromised?

The experiment with the Rab11 mutant is interesting but it addresses the role of Rab11 in the recruitment of NP to spiky ICVs, not the role of the ER in their formation.

Our study opens new avenues for future research on the role of the ER in the formation of ICVs, as mentioned in the discussion, page 16 lines 9-21. In particular, we point to our observation that a particular form of the ER markers PDI and calreticulin accumulate in the fraction of IAV-infected cells that is enriched in ICVs, with an apparent molecular weight which differs from that detected in mock-infected cells. These findings suggest that IAV infection could trigger post-translational modifications of ER proteins, which in turn could mediate remodelling of the endomembrane system and the formation of ICVs (Fig. 9, page 13 lines 12-18, discussed page 16 lines 12-21).

Therefore, the first conclusion of this article is that virus RNPs (marked by NP) are associated to 100-200nm Rab11 positive structures that could (or not) be small endosomes. This has not been addressed in any way. No endosomal markers in addition to Rab11 has been used to attempt to identify ICVs.

In the revised version of the manuscript, new immunoEM data are provided which clearly highlight differences between recycling endosomes and ICVs. Recycling endosomes show Rab11 and transferrin receptor labeling and an electron-dense lumen. By contrast, ICVs show Rab11 signal but are almost devoid of transferrin receptor signal, and they show an electron-lucent lumen (Supp. Fig. 14, page 11 lines 5-11).

In addition, there is no attempt to characterize the spiky coat. Can the authors at least rule out the involvement of clathrin. I am not saying that the entire coat is clathrin but it would be present there!

In the revised version of the manuscript, we have included an additional Supplementary figure to show that the irregular coat of ICVs differs from the well described cellular coats involved in transport between organelles, such as clathrin and caveolin (Supp. Fig. 2, page 6 lines 23-25). Consistently, clathrin was not detected in the subcellular fraction from infected cells that was enriched in ICVs (Fig. 9, page 13 lines 10-12).

We have also further characterized the spiky coat of ICVs:

(i) we have measured the length of 137 filaments from 16 ICVs (Supp. Fig. 8); it ranged from 26.6 to 112.5 nm, which is consistent with the 30-110 nm range previously described for vRNPs (stated page 8 lines 23-26).

(ii) we have performed PB2 + NP double-labeling by immuno-EM: ICVs showed signal for both NP and PB2, strengthening our conclusion that they are coated with vRNPs (Supp. Fig. 11, page 9 lines 26-27 and page 10 lines 1-4).

(iii) we performed negative staining of an aliquot of the subcellular fraction enriched in ICVs: the irregular coat shows the same morphology as isolated vRNPs (Sugita et al, J Virol 2013) (Fig. 9B and page 13 lines 1-3).

Taken altogether, these observations clearly demonstrate that the spiked filaments at the surface of ICVs are vRNPs.

The second conclusion of the paper is that the ER is extensively remodeled and that the ribosomes cluster. Whether it has to do with the ICVs formation remains to be demonstrated. It is also (likely) possible that the ribosome clustering leads to the ER tubulation. The relationship between ribosome presence on the ER and its sheet/tubule morphology has been addressed in an old paper from the Jokitalo's group.

Is there a relationship between ribosome clustering and infection, formation of ICVs. Could the experiment in Figure 2 (kinetics) be quantified to inform on the link between these phenotypes? The quantification of the three phenotypes (accumulation of ICVs near MTOC / ER remodeling / ribosome clusters) is presented in the Supp. Table 2 of the revised version of the manuscript. The data show that all three phenotypes are induced by IAV infection, and suggest that they could be related to each other. These observations are mentioned page 7 lines 10-14 of the revised version of the manuscript.

We thank the reviewer for drawing our attention to the article by Jokitalo et al. It is cited in the revised version of the manuscript (page 14 lines 24-26, reference #36).

Minor

-Add indication on the figures, gold size corresponding to what, time of infection, color coding for the IF etc. It will make the reading much easier.

This request has been fulfilled.

-Rab11 labeling seems to decrease at the peripheral region

METTEM images provided in Supp. Fig. 6 (re-numbered Supp. Fig. 13 in the revised version) show that in mock-infected cells, Rab11 is detected in vesicles and sporadically also on ER cisternae. In contrast, in IAV-infected cells, a strong Rab11 signal can be detected on large portions of the remodeled ER (Fig. 4A, re-numbered Fig. 6A in the revised version), suggesting that that IAV infection triggers the relocation and/or redistribution of Rab11 proteins in the ER. These observations are mentioned page 10 lines 25-27 and page 11 line 1, and discussed page 16 lines 6-12 of the revised version of the manuscript.

-deconvolute 2E to appreciate the strong remodeling of the ER upon infection

Figures 2E-2F (renumbered 3E-3F in the revised version) have not been deconvoluted in order to preserve the large size of the images shown in Fig. 3.

REVIEWERS' COMMENTS:

Reviewer #1 (Remarks to the Author):

In response to reviewers' comments, Castro Martin et al. have addressed most questions raised using quantification, biochemical approaches, and providing additional EM images. They now show more convincingly that ICVs are what they claim to be and that they are vRNP-carriers associated to the ER and Rab11. Quantification shows that ER-associated ICVs are predominant compared to cytoskeletal filament-associated ones. ICVs could be enriched by gradient centrifugation, indicating the presence of vRNP-like protrusions from their surface by EM. IAV-induced modifications of ER-associated proteins are observed, which suggests that the virus remodels the ER in order to facilitate viral egress. Their model implies that rather than depending on microtubule motors and Rab11-positive recycling endosomes, the neo-vRNPs directly associate with the external nuclear membrane and the ER, following nuclear export from nuclear pore complexes. This mode sounds more efficient - the pathway is continuous from the NPC to the PM, and can bypass the MTOC. (Live imaging of ER morphology during viral replication will provide more hints as how the remodelling takes place, though this is not the scope of this study.) Finally, the precise touch-and-go mechanism by which the ICVs-bound vRNPs are transferred to the PM remains to be solved. Overall this is a very interesting study that sheds light on viral egress pathways and virus-host interactions.

Reviewer #2 (Remarks to the Author):

From my point of view the new data (double labelling and organelle fractionation experiments as well as comprehensive quantification of EM-pictures) eliminated my doubts regarding the major claims of the paper.

I have just two minor points:

Minor comments:

1. Page 8, line 14-15: "Taken together, our data demonstrate that viral membrane proteins are not requested for ER remodeling and ICV formation."

It is better to write "transmembrane protein" since M1 was not excluded. Actually, M1 might do the job since it (also) localizes to the ER and has membrane remodelling features.

2. Page 16, line 15:

"In the case of PDI, this difference (in the molecular weight) could correspond to changes in the redox state of the protein, ..."

It is probably known (i. e. published) whether changes in the redox state of PDI can cause a (rather large) change in its SDS-PAGE mobility. At first glance it seems unlikely to me.

Reviewer #3 (Remarks to the Author):

I see that the authors have invested a lot of work into revising the manuscript according to the comments of all reviewers. They addressed almost all my comments, except the IF of Rab11 that at the ER. I understand that this experiment might be technically challenging. In light of the excellent EM images, I am happy with the revised version and have no further remarks.

Reviewer #4 (Remarks to the Author):

The authors have come a long way to answer the many but convergent comments of the reviewers, including mine.

The quality of the electron micrographs is much better. The story is better documented. The quantitative aspect of the phenotype as well as the demonstration that ICVs are not recycling endosomes are both very positive in improving the manuscript.

1) I particularly appreciate the demonstration that in 20 cells, the three phenotypes go hand in hand: ER remodelling, ribosome clustering and ICVs accumulation. This indeed suggests that the remodelled ER (that becomes Rab11 positive) is involved in the ICV biogenesis.

I would however like the authors to state whether these cells were selected randomly and whether those were not among many other cells that do not show this convergence in phenotype.

2) Furthermore, I still urge the authors to refrain from saying that ICVs bud from the ER. If this were that obvious, the beautiful tomogram (Figure 8) would show some membrane continuity between the ER and the ICVs and the budding aspect would become apparent.

Of note, the profiles shown by the authors in Suppl 15 and 16 are suggestive of this budding but does not prove it. For the first one, the ER aspect is very small (few ribosomes) and for the second, the membrane continuity is not obvious. These are also the only two events shown in the whole paper, so although it is possible that ICVs can bud from the ER, the authors need to acknowledge that these events are rare and that the remodelled ER could contribute in other way to the ICV biogenesis.

The authors acknowledge this (as they did in the rebuttal in the response to my queries on this, point 7) and refrain from saying that ICV bud from the ER in a definitive strong affirmative tone. This is counter productive.

-The model/scheme at the end of the paper (Figure 10) should be re-drawn to remove the many budding profiles of ICVs from the ER. This is misleading and NOT in line with the pictures provided in the ms where most of the ICVs are not even close to ER.

-Also change the sentence in the abstract "and on irregularly coated vesicles (ICVs) that bud from the ER and touch the plasma membrane".

-Line 18 page 5 introduction: Change the sentence "We provide evidence that Rab11 function is required for ICV formation, ICVs bud from the ER, and they touch the plasma membrane".

Please, rephrase along the lines that "the remodelled ER is involved in ICV biogenesis, from which they perhaps bud"

Also modify the other places in the ms where this is mentioned.

3) Same thing for "touch the plasma membrane" or "Direct contact with plasma membrane". This is NOT shown in the figure and the authors mention smaller than 200nm which is by no means a direct contact. Again, in the whole paper, hardly any ICVs are in direct contact with the PM.

Specific comments:

4) I would think that the IEM showing the difference between recycling endosomes and the WB showing these differences biochemically (Figure 9) should be combined and make the bulk of a new paragraph. This difference is a very important piece of information.

Also on Suppl Figures

5) Do mark the panels of the figures for the labelling and the cell manipulation especially the IEM upon Rab11 and Rab11 DN expression. There are many many figures to this paper (and suppl materials) and this direct information will avoid many back and forth between different parts of the

text and the figures. I insist on this.

6) I think there is a mix up in suppl Fig14. The big gold is meant to be Tfr and the small Rab11 and it seems to me that the ICVs are decorated by big gold, not small. Again, this would be way simpler if the figure was labelled as I request in my point 4.

Once this is done, the ms should be suitable for publication.

Response to reviewers of Manuscript NCOMMS-17-01013A

« A new class of Rab11-dependent vesicles transports influenza virus genome from a modified endoplasmic reticulum to the plasma membrane »

We would like to thank the reviewers for their additional comments and suggestions. The text and legends of figures have been modified in order to address their remaining concerns and to comply with the editorial requests. A revised version of the manuscript with highlighted changes is provided.

Below is a point-by-point response to each of the reviewers' comments. The page and line numbers refer to the new version of the manuscript.

REVIEWERS' COMMENTS:

Reviewer #1 (Remarks to the Author):

In response to reviewers' comments, Castro Martin et al. have addressed most questions raised using quantification, biochemical approaches, and providing additional EM images. They now show more convincingly that ICVs are what they claim to be and that they are vRNP-carriers associated to the ER and Rab11. Quantification shows that ER-associated ICVs are predominant compared to cytoskeletal filament-associated ones. ICVs could be enriched by gradient centrifugation, indicating the presence of vRNP-like protrusions from their surface by EM. IAV-induced modifications of ER-associated proteins are observed, which suggests that the virus remodels the ER in order to facilitate viral egress. Their model implies that rather than depending on microtubule motors and Rab11-positive recycling endosomes, the neo-vRNPs directly associate with the external nuclear membrane and the ER, following nuclear export from nuclear pore complexes. This mode sounds more efficient - the pathway is continuous from the NPC to the PM, and can bypass the MTOC. (Live imaging of ER morphology during viral replication will provide more hints as how the remodelling takes place, though this is not the scope of this study.) Finally, the precise touch-and-go mechanism by which the ICVs-bound vRNPs are transferred to the PM remains to be solved. Overall this is a very interesting study that sheds light on viral egress pathways and virus-host interactions.

Reviewer #2 (Remarks to the Author):

From my point of view the new data (double labelling and organelle fractionation experiments as well as comprehensive quantification of EM-pictures) eliminated my doubts regarding the major claims of the paper.

I have just two minor points:

Minor comments:

1. Page 8, line 14-15: "Taken together, our data demonstrate that viral membrane proteins are not requested for ER remodeling and ICV formation."

It is better to write "transmembrane protein" since M1 was not excluded. Actually, M1 might do the job since it (also) localizes to the ER and has membrane remodelling features.

We agree with the reviewer's comment and have modified the sentence accordingly (page 8 line 15 of the revised version of the manuscript)

2. Page 16, line 15:

"In the case of PDI, this difference (in the molecular weight) could correspond to changes in the redox state of the protein, ..."

It is probably known (i. e. published) whether changes in the redox state of PDI can cause a (rather

large) change in its SDS-PAGE mobility. At first glance it seems unlikely to me. Indeed, it has been published that changes in the redox state of PDI can cause a substantial change in its SDS-PAGE mobility, as for instance in the article by Zhang et al, *Different Interaction Modes for Protein-disulfide Isomerase (PDI) as an Efficient Regulator and a Specific Substrate of Endoplasmic Reticulum Oxidoreductin-1 α (Ero1 α)*, 2014, *The Journal of Biological Chemistry* 289, 31188–31199. This reference (#39) was included in the the revised version of the manuscript, page 16 line 17.

Reviewer #3 (Remarks to the Author):

I see that the authors have invested a lot of work into revising the manuscript according to the comments of all reviewers. They addressed almost all my comments, except the IF of Rab11 that at the ER. I understand that this experiment might be technically challenging. In light of the excellent EM images, I am happy with the revised version and have no further remarks.

Reviewer #4 (Remarks to the Author):

The authors have come a long way to answer the many but convergent comments of the reviewers, including mine.

The quality of the electron micrographs is much better. The story is better documented. The quantitative aspect of the phenotype as well as the demonstration that ICVs are not recycling endosomes are both very positive in improving the manuscript.

1) I particularly appreciate the demonstration that in 20 cells, the three phenotypes go hands in hands: ER remodelling, ribosome clustering and ICVs accumulation. This indeed suggests that the remodelled ER (that becomes Rab11 positive) is involved in the ICV biogenesis.

I would however like the authors to state whether these cells were selected randomly and whether those were not among many other cells that do not show this convergence in phenotype.

The cells were selected randomly. This information was included in the the revised version of the manuscript, page 7 line 12.

2) Furthermore, I still urge the authors to refrain from saying that ICVs bud from the ER.

If this were that obvious, the beautiful tomogram (Figure 8) would show some membrane continuity between the ER and the ICVs and the budding aspect would become apparent.

Of note, the profiles shown by the authors in Suppl 15 and 16 are suggestive of this budding but does not prove it. For the first one, the ER aspect is very small (few ribosomes) and for the second, the membrane continuity is not obvious. These are also the only two events shown in the whole paper, so although it is possible that ICVs can bud for the ER, the authors need to acknowledge that these events are rare and that the remodelled ER could contribute in other way to the ICV biogenesis.

The authors acknowledge this (as they did in the rebuttal in the response to my queries on this, point 7) and refrain from saying that ICV bud from the ER in a definitive strong affirmative tone. This is counter productive.

-The model/scheme at the end of the paper (Figure 10) should be re-drawn to remove the many budding profiles of ICVs from the ER. This is misleading and NOT in line with the pictures provided in

the ms where most of the ICVs are not even close to ER.

The schematic model proposed Figure 10 has been redrawn. A single budding profile of ICV from the ER is now shown, to suggest the possibility that ICV could bud from the ER. The other schematic ICVs were positioned at various distances from the ER membrane.

-Also change the sentence the abstract “and on irregularly coated vesicles (ICVs) that bud from the ER and touch the plasma membrane”.

The sentence has been changed to “Some ICVs are found very close to the ER and to the plasma membrane.”

-Line 18 page 5 introduction: Change the sentence “We provide evidence that Rab11 function is required for ICV formation, ICVs bud from the ER, and they touch the plasma membrane”.

Please, rephrase along the lines that “the remodelled ER is involved in ICV biogenesis, from which they perhaps bud”

The sentence has been changed to “Some ICVs are found very close to the ER and to the plasma membrane, and we provide evidence that Rab11 function is required for ICV formation. Overall our data strongly support a model in which (i) the modified ER is the first station of vRNPs after their exit from the nucleus, (ii) the ER is involved in the Rab11-dependent biogenesis of ICVs displaying Rab11 and vRNPs (iii) ICVs then serve as the transport organelle for vRNPs from the ER to the plasma membrane.” (page 5 lines 17-22).

Also modify the other places in the ms where this is mentioned.

The sentences that mentioned budding of ICVs from the ER have been rephrased along the lines suggested by the reviewer, e.g. page 11 line 25, page 14 line 11, page 15 lines 26-27, page 35 line 14 of the revised version of the manuscript.

3) Same thing for “touch the plasma membrane” or “Direct contact with plasma membrane”. This is NOT shown in the figure and the authors mention smaller than 200nm which is by no mean a direct contact. Again, in the whole paper, hardly any ICVs are in direct contact with the PM.

The sentences that mentioned direct contact of ICVs with the plasma membrane have been rephrased along the lines suggested by the reviewer, e.g. page 11 line 25, page 12 lines 7-8, page 17 lines 26-27, page 35 line 18 of the revised version of the manuscript.

Specific comments:

4) I would think that the IEM showing the difference between recycling endosomes and the WB showing these differences biochemically (Figure 9) should be combined and make the bulk of a new paragraph. This difference is a very important piece of information.

The fact that the results obtained upon biochemical analysis and IEM are consistent and altogether strengthen the conclusion that ICVs are distinct from recycling endosomes has been underlined in the revised version of the manuscript (page 13 lines 9-10).

5) Do mark the panels of the figures for the labelling and the cell manipulation especially the IEM upon Rab11 and Rab11 DN expression. There are many many figures to this paper (and suppl materials) and this direct information will avoid many back of forth between different parts of the text and the figures. I insist on this.

The panels of the figures showing IEM data (Fig. 7, Suppl. Fig. 11 and Suppl. Fig. 14) have been labelled as requested by the reviewer.

6) I think there is a mix up in suppl Fig14. The big gold is meant to be Tfr and the small Rab11 and it seems to me that the ICVs are decorated by big gold, not small. Again, this would be way simpler if

the figure was labelled as I request in my point 4.

The double labelling in the supplementary figure 14 was done sequentially according to the protocol using the glutaraldehyde block between antibodies as published by Slot et al. (1991) (Reference #56 in the new version, Reference #53 in the previous version of the manuscript. For this double labelling we had to use the mouse anti TfR antibody (clone 68.1, thermofisher 13-6800) first because it is a mouse IgG1 and we use a rabbit anti mouse bridging antibody (Dako Z0259) that shows a small cross reaction with rat IgGs. Because the antibody against HA is a rat IgG1 (clone 3F10, Roche, 11867423001) we had to use this antibody in the second step of the labelling, which is possible because the rabbit anti rat bridging antibody (epitomics 3030-1, R18-2) does not crossreact with mouse IgGs. In our hands double labellings work always better when for the first antibody the smaller protein A gold size is used (here 10 nm) and for the second antibody the larger protein A gold (here 15 nm) is used.

The double labelling presented in supplementary figure 14 differs from the one presented in Figure 7 of the manuscript. In Figure 7 a double labelling is done with the rat anti HA antibody, using the bridging antibody followed by protein A gold (5 nm). The second antibody is the mouse anti NP (clone HT103 Kerafast, #EMS101), which is an IgG2a and reacts directly with protein A gold (here 10 nm) (see Griffiths, Fine structure Immunocytochemistry, 1993, chapter 6, page 224). For this reason we could use here the HA antibody in the first step.

Once this is done, the ms should be suitable for publication.